# Comparing Causal Frameworks: Potential Outcomes, Structural Models, Graphs, and Abstractions

**Duligur Ibeling**
Department of Computer Science
Stanford University
duligur@stanford.edu

**Thomas Icard**
Department of Philosophy
Stanford University
icard@stanford.edu

## Abstract

The aim of this paper is to make clear and precise the relationship between the Rubin causal model (RCM) and structural causal model (SCM) frameworks for causal inference. Adopting a neutral logical perspective, and drawing on previous work, we show what is required for an RCM to be *representable* by an SCM. A key result then shows that every RCM—including those that violate algebraic principles implied by the SCM framework—emerges as an abstraction of some representable RCM. Finally, we illustrate the power of this conciliatory perspective by pinpointing an important role for SCM principles in classic applications of RCMs; conversely, we offer a characterization of the algebraic constraints implied by a graph, helping to substantiate further comparisons between the two frameworks.

Toward the end of the twentieth century several frameworks arose for formalizing and analyzing problems of causal inference. One of these, associated with Rubin [34] and others (see [16]), takes the *potential outcome*—formalizing the effect of one variable on another—as a fundamental target of analysis. Causal assumptions in the Rubin causal model (RCM) are naturally encoded as algebraic constraints on potential outcomes, and research in this area has spawned a remarkable body of theoretical and applied work especially in social and biomedical sciences (see [14] for a review).

A second approach, associated with Pearl [26] and collaborators (see [27] for a textbook treatment; see also [37]), focuses instead on assumptions that can be encoded qualitatively, or more specifically, graphically, arising from a fundamental object known as a structural causal model (SCM). The *do-calculus* is one of the crowning achievements of this approach, and it has been shown derivationally complete with respect to a wide range of canonical estimation and inference problems [35, 2, 19].

Both frameworks have enjoyed considerable influence within recent causal machine learning in particular. As just one example, concern in reinforcement learning about the possibility of unobserved confounders—variables impacting both decisions and their outcomes—has generated a number of important advances, some employing tools and concepts from the RCM approach (e.g., [18, 25, 17]), others grounded in the SCM approach and typically involving graphs (e.g., [3, 38, 42]).

Despite the remarkable successes that both of these frameworks have engendered in machine learning and beyond, there remains substantial controversy over how to understand their relationship. In the literature it has been claimed, on the one hand, that the two are equivalent, that "a theorem in one is a theorem in the other" [27, p. 98]. On the other hand, some authors suggest that the two are only equivalent in a weak sense, one that "builds the limitations of SCM into the resulting comparison and likewise filters out aspects of the rival theory that do not readily translate to SCM" [21, p. 443].

At issue are two separable questions. The first is one of practical significance. Some argue that graphs give greater "conceptual clarity" [27, p. 103] and that SCMs more generally offer a "a flexible formalism for data-generating models" that helps ground causal inquiry [4, p. 514]; others argue that work in the RCM framework provides "transparent definitions of causal effects and encourages the analyst to consider individual-level heterogeneity as a first principle" [23, p. 91] as well as "guidance

37th Conference on Neural Information Processing Systems (NeurIPS 2023).

to researchers and policy makers for practical implementation" [14, p. 1131]. While obviously very important, our goal is not to address these disputes about what theoretical primitives are most "natural" or "useful" for practitioners or applied researchers. Rather, the aim of the present contribution is to offer a number of new technical results that together shed light on a more basic question, namely, how precisely the RCM and SCM frameworks relate at a theoretical level. For example, are the two merely notational variants, or does one tacitly enforce assumptions that the other does not?

In this paper we first endeavor, building on previous work, to elucidate the precise sense in which SCMs can be construed as *representations* of RCMs, provided the latter satisfy two key principles known as composition and reversibility [9, 10]. Interestingly, such principles (or their logical consequences) have been questioned in the literature (e.g., [7]). Our second goal is to help clarify the sense in which they may fail. Drawing from recent literature on causal abstraction (e.g., [33, 5])— broadened to cover both SCMs and RCMs—we suggest that failure of these principles plausibly results when causally relevant low-level details are elided in favor of more abstract variables. Our Thm. 1 buttresses this intuition, showing that every RCM is a *constructive abstraction* of a representable RCM (hence satisfying composition and reversibility). We furthermore remark on how the well known SUTVA assumptions [16] can be understood as conditions on good variable abstractions.

Our starting point in this work is theoretically neutral, taking for granted only the primitive, "probability of a counterfactual." In the second half of the paper we introduce a framework-neutral formal language for reasoning about such probabilities, which in turn facilities further comparison. With respect to this common language, we offer a completeness result for the class of all RCMs (Thm. 2) and, drawing on [10], the class of representable RCMs (Cor. 2). These results are illustrated with an example derivation of LATE (see [15, 1]), which also helps illuminate which assumptions are logically required in the derivation. Meanwhile, we offer a partial answer to the well-known open question [39, 31] of how to characterize the algebraic constraints implied by a particular graph (Thm. 3), a result that helps bring graphical assumptions into this neutral common language. Finally, we show how an existing result on single-world intervention graphs (SWIGs), a framework drawing from both perspectives, can be construed as a completeness result for the same language (Thm. 4).

Taken together, our results are largely conciliatory—in the same spirit as other important conciliatory work in this context; see, e.g., [30, 36, 20]—showing how the two frameworks are productively compatible, while also suggesting distinctive perspectives on problems of causal inference.

Proofs are deferred to supplementary appendices A, B, which contain additional technical material.

# 1 Modeling

We first introduce a formalization of the Rubin causal model [34, 11, 12] before then turning to structural causal models [26, 27, 4]. The relationship between these two is elucidated in §1.1.3.

## 1.1 Preliminaries

Common to both frameworks will be a set $\mathbf{V}$ of *endogenous variables*. Concerning notation:

**Notation.** The signature (or range) of a variable $V$ is denoted $\mathrm{Val}(V)$. Where $\mathbf{S}$ is a set of variables, let $\mathrm{Val}(\mathbf{S}) = \bigtimes_{S \in \mathbf{S}} \mathrm{Val}(S)$, the product of the family of sets $\mathrm{Val}(S)$ indexed by $S \in \mathbf{S}$. Elements of $\mathrm{Val}(\mathbf{S})$ represent joint valuations of the variables $\mathbf{S}$. Given an indexed family of sets $\{S_\beta\}_{\beta \in B}$ and elements $s_\beta \in S_\beta$, let $\{s_\beta\}_\beta$ denote the indexed family of elements whose object associated with the index $\beta$ is $s_\beta$, for all $\beta$. The symbol $\subset$ indicates any subset (or set inclusion) and does not imply a strict subset (or proper inclusion). For $B' \subset B$ write $\pi_{B'} : \bigtimes_{\beta \in B} S_\beta \to \bigtimes_{\beta \in B'} S_\beta$ for the *projection map* sending each $\{s_\beta\}_{\beta \in B} \mapsto \{s_{\beta'}\}_{\beta' \in B'}$; abbreviate $\pi_{\beta'} = \pi_{\{\beta'\}}$, for $\beta' \in B$. Thus if $\mathbf{s} \in \mathrm{Val}(\mathbf{S})$ is a joint valuation of variables $\mathbf{S}$ and $S \in \mathbf{S}$ is a single variable, then $s = \pi_S(\mathbf{s}) \in \mathrm{Val}(S)$ is a value for $S$. If $\mathbf{S}' \subset \mathbf{S}$ then $\pi_{\mathbf{S}'}(\mathbf{s}) \in \mathrm{Val}(\mathbf{S}')$ is a joint valuation of $\mathbf{S}'$, namely the projection of $\mathbf{s}$ to $\mathbf{S}'$. Upper-case letters like $\mathbf{S}$ conventionally represent those sets of variables that the corresponding lower-case letters $\mathbf{s}$ are valuations of, $\mathbf{s} \in \mathrm{Val}(\mathbf{S})$.

### 1.1.1 Rubin Causal Models, Potential Outcomes, Counterfactuals

The present formalization of the Rubin causal model [34, 16] loosely follows previous presentations; see especially [12]. It codifies experimental outcomes across individuals drawn from a distribution.

These are *potential outcomes* over a variable set $\mathbf{V}$, defined as expressions of the form $Y_{\mathbf{x}}$ for an *outcome* $Y \in \mathbf{V}$ and an *intervention* or *treatment* $\mathbf{x} \in \mathrm{Val}(\mathbf{X})$ for some $\mathbf{X} \subset \mathbf{V}$, and interpreted as the value observed for $Y$ in a controlled experiment where each $X \in \mathbf{X}$ is held fixed to $\pi_X(\mathbf{x})$.

**Definition 1.** A *Rubin causal model* (RCM) is a tuple $\mathcal{R} = \langle \mathcal{U}, \mathbf{V}, \mathfrak{O}, \mathfrak{F}, P \rangle$ where $\mathcal{U}$ is a finite set of *units* or *individuals*, $\mathbf{V}$ is a finite set of *endogenous variables*, $\mathfrak{O}$ is a set of potential outcomes over $\mathbf{V}$, $\mathfrak{F}$ is a set of *potential response* functions, to be defined shortly, and $P : \mathcal{U} \to [0,1]$ is a probability distribution on $\mathcal{U}$. A potential response for $Y_{\mathbf{x}} \in \mathfrak{O}$ is a function $f_{Y_{\mathbf{x}}} : \mathcal{U} \to \mathrm{Val}(Y)$. For each $Y_{\mathbf{x}} \in \mathfrak{O}$ we require that $\mathfrak{F} = \{f_{Y_{\mathbf{x}}}\}_{Y_{\mathbf{x}} \in \mathfrak{O}}$ contain exactly one such function.[1]

RCMs are often specified in a tabular form as in, e.g., Fig. 1 below. We adopt the notation $y_{\mathbf{x}}(i)$ or $Y_{\mathbf{x}}(i) = y$ as a shorthand for $f_{Y_{\mathbf{x}}}(i) = y$: for $\mathcal{R}$ as in Def. 1, write $\mathcal{R} \vDash y_{\mathbf{x}}(i)$ iff $Y_{\mathbf{x}} \in \mathfrak{O}$ and $f_{Y_{\mathbf{x}}}(i) = y$. This means that in the above controlled experiment, outcome $y \in \mathrm{Val}(Y)$ is observed for individual $i$. Each $Y_{\mathbf{x}}$ can be thought of as a variable with range $\mathrm{Val}(Y_{\mathbf{x}}) = \mathrm{Val}(Y)$. We call the set $\mathrm{Val}(\mathfrak{O})$ of joint valuations of these variables *counterfactuals*. A set of potential responses $\mathfrak{F}$ then maps units to counterfactuals, $\mathfrak{F} : \mathcal{U} \to \mathrm{Val}(\mathfrak{O})$, by defining $\mathfrak{F}(i) = \{f_{Y_{\mathbf{x}}}(i)\}_{Y_{\mathbf{x}} \in \mathfrak{O}}$, and:

**Definition 2.** Where $\mathcal{R}$ is as in Def. 1, the *counterfactual distribution* $P_{\mathrm{cf}}^{\mathcal{R}} : \mathrm{Val}(\mathfrak{O}) \to [0,1]$ induced by $\mathcal{R}$ is the pushforward[2] $\mathfrak{F}_*(P)$ of $P : \mathcal{U} \to [0,1]$ under $\mathfrak{F} : \mathcal{U} \to \mathrm{Val}(\mathfrak{O})$.

The reason we call $P_{\mathrm{cf}}^{\mathcal{R}}$ a counterfactual distribution (and $\mathrm{Val}(\mathfrak{O})$ counterfactuals) is because such joint probabilities over multiple potential outcomes appear in the usual ratio definition of counterfactual probabilities. For instance, $P(y_x | y'_{x'}) = P(y_x, y'_{x'})/P(y'_{x'})$ gives the probability that a unit who was withheld treatment and did not recover would have recovered if assigned treatment. But $P_{\mathrm{cf}}^{\mathcal{R}}$ also answers via marginalization all questions (whenever defined by $\mathcal{R}$) about "interventional" probabilities like $P(y_{\mathbf{x}})$, as well as purely "observational" probabilities such as $P(\mathbf{x})$; see, e.g., [4].

Some authors submit that "probability will mean nothing more nor less than a proportion of units" [11, p. 945], thereby assuming a uniform distribution on a finite population $\mathcal{U}$ (cf. also [1]). Of course, in the infinite population size limit we recover all RCMs as in Def. 1 (see Prop. A.1).

While practitioners do not typically consider potential outcomes $Y_{\mathbf{x}}$ when $Y \in \mathbf{X}$, instead maintaining a strict dichotomy between cause and effect variables (e.g., [11, 12]), it is natural to impose the following requirement (known as *effectiveness*) whenever such potential outcomes are defined. An intervention is always assumed to be a *successful* intervention: whenever $Y \in \mathbf{X}$,

$$\text{Effectiveness.} \qquad Y_{\mathbf{x}}(u) = \pi_Y(\mathbf{x}) \qquad (1)$$

for every $u \in \mathcal{U}$. In fact, practice in the RCM framework reflects this same assumption, in the sense that violations of it are taken to signify a poor choice of variables. As a classic example, the possibility of *non-compliance* in an experimental trial motivates the introduction of instrumental variables, and specifically a separation between, e.g., *treatment* and *assignment to treatment* (cf. Ex. 3). Crucially, we recover effectiveness with respect to the latter. We will assume any RCM to meet (1) unless otherwise specified; let $\mathfrak{R}_{\mathrm{eff}}$ be the class of such RCMs.[3]

### 1.1.2 Structural Causal Models

An important feature of RCMs is that potential outcomes are cleanly separated from assignment mechanisms [16]. A different starting point is to assume that potential outcomes and their algebraic behavior are rather *derived* from an underlying formal representation of causal structure. These putatively "deeper mathematical objects" [27, p. 102] involve concrete functional dependencies, and an operation of function replacement known as *intervention*:

**Definition 3.** A *structural causal model* (SCM) is a tuple $\mathcal{M} = \langle \mathbf{U}, \mathbf{V}, \mathcal{F}, P(\mathbf{U}) \rangle$ where $\mathbf{U}$ is a finite set of *exogenous variables*, $\mathbf{V}$ is a finite set of *endogenous variables*, $\mathcal{F}$ is a family of *structural*

---

[1] By generalizing to allow multiple such functions one arrives at a class of models closely related to the *generalized structural equation models* (GSEMs) of Peters and Halpern [29], or the equivalent class of *causal constraint models* (CCMs) introduced by Blom et al. [6]. Rather than mapping each potential outcome $Y_{\mathbf{x}}$ to a value in $\mathrm{Val}(Y)$, GSEMs map each (allowable) intervention $\mathbf{x}$ to a (possibly empty) *set* of values for all the variables, that is, to elements of the powerset $\wp(\mathrm{Val}(\mathbf{V}))$. RCMs, by contrast, allow that, e.g., $Y_{\mathbf{x}}$ may be defined for all $\mathbf{u}$, while $Z_{\mathbf{x}}$ is undefined because $Z_{\mathbf{x}} \notin \mathfrak{O}$. The two are thus incomparable in expressive power.

[2] That is, $P_{\mathrm{cf}}^{\mathcal{R}}(\mathbf{o}) = P(\mathfrak{F}^{-1}(\{\mathbf{o}\}))$ for any $\mathbf{o} \in \mathrm{Val}(\mathfrak{O})$.

[3] Not only does this assumption reflect practice, but it is also without loss with regard to comparing RCMs and SCMs, as the latter also satisfy effectiveness: see footnote 4.

*functions*, to be defined shortly, and $P : \mathrm{Val}(\mathbf{U}) \to [0, 1]$ is a probability distribution on (joint valuations of) $\mathbf{U}$.

A structural function for $V \in \mathbf{V}$ is a function $f_V : \mathrm{Val}(\mathbf{U}_V \cup \mathbf{Pa}_V) \to \mathrm{Val}(V)$, where $\mathbf{U}_V \subset \mathbf{U}$, $\mathbf{Pa}_V \subset \mathbf{V} \setminus \{V\}$. For every $V \in \mathbf{V}$ we require that $\mathcal{F} = \{f_V\}_V$ have exactly one such function; the entire collection $\mathcal{F}$ thus forms an exogenous-to-endogenous mapping.

Interventions come as a derived notion, replacing structural functions with constant functions [22, 27]:

**Definition 4** (Intervention on SCMs). Let $\mathbf{x} \in \mathrm{Val}(\mathbf{X})$ for some $\mathbf{X} \subset \mathbf{V}$ be an intervention and $\mathcal{M}$ be the SCM of Def. 3. Then define a *manipulated model* $\mathcal{M}_\mathbf{x} = \langle \mathbf{U}, \mathbf{V}, \{f'_V\}_V, P(\mathbf{U})\rangle$ where each $f'_V : \mathrm{Val}(\mathbf{U}'_V \cup \mathbf{Pa}'_V) \to \mathrm{Val}(V)$. If $V \notin \mathbf{X}$ define $\mathbf{U}'_V = \mathbf{U}_V$, $\mathbf{Pa}'_V = \mathbf{Pa}_V$, and $f'_V = f_V$. If $V \in \mathbf{X}$ define $\mathbf{U}'_V = \mathbf{Pa}'_V = \varnothing$ and $f'_V$ as a constant function mapping to $\pi_V(\mathbf{x})$.

Letting $\mathcal{M}$ be the SCM of Def. 3, for $\mathbf{v} \in \mathrm{Val}(\mathbf{V})$ and $\mathbf{u} \in \mathrm{Val}(\mathbf{U})$ write $\mathcal{M}, \mathbf{u} \vDash \mathbf{v}$ if we have $f_V\big(\pi_{\mathbf{U}_V \cup \mathbf{Pa}_V}(\mathbf{v})\big) = \pi_V(\mathbf{v})$ for every $V$. Let $\mathfrak{M}_{\mathrm{uniq}}$ be the class of all SCMs $\mathcal{M}$ such that, for any $\mathbf{u}$ and intervention $\mathbf{x}$ there is a unique *solution* $\mathbf{v}$ such that $\mathcal{M}_\mathbf{x}, \mathbf{u} \vDash \mathbf{v}$. In this case we define the potential outcome $Y_\mathbf{x}(\mathbf{u})$ as $\pi_Y(\mathbf{v})$. Thus any $\mathcal{M} \in \mathfrak{M}_{\mathrm{uniq}}$ defines a potential outcome for *every* $Y_\mathbf{x}$, giving a natural function $p^\mathcal{M} : \mathrm{Val}(\mathbf{U}) \to \mathrm{Val}(\{Y_\mathbf{x}\}_{\mathrm{all}\, Y, \mathbf{x}})$ via these outcomes, and:

**Definition 5.** The counterfactual distribution $P_{\mathrm{cf}}^\mathcal{M} : \mathrm{Val}(\{Y_\mathbf{x}\}_{\mathrm{all}\, Y, \mathbf{x}}) \to [0, 1]$ induced by $\mathcal{M} \in \mathfrak{M}_{\mathrm{uniq}}$ is the pushforward $p_*^\mathcal{M}(P)$ of $P : \mathrm{Val}(\mathbf{U}) \to [0, 1]$ under $p^\mathcal{M} : \mathrm{Val}(\mathbf{U}) \to \mathrm{Val}(\{Y_\mathbf{x}\}_{\mathrm{all}\, Y, \mathbf{x}})$.

Thus, SCMs in $\mathfrak{M}_{\mathrm{uniq}}$ canonically define counterfactual distributions for all possible potential outcomes via manipulation of functional dependencies. Importantly, Def. 5 provides a bridge to RCMs, as both produce counterfactual distributions (recall Def. 2). As long as the counterfactual probabilities are assumed to mean the same thing—i.e., as long as they highlight the same targets for empirical and theoretical investigation—we can then compare the ranges of assumptions and inference patterns that each framework can encode about them. We thus assume that all our SCMs belong to this class $\mathfrak{M}_{\mathrm{uniq}}$.

### 1.1.3 Representation of RCMs by SCMs

A contentious methodological question is whether all (endogenous) variables should be potential candidates for intervention. Following the literature we have supposed that SCMs allow all possible interventions (though this assumption is not universal; see, e.g., [33, 5]). For RCMs it is generally assumed that there can be "no causation without manipulation" [11, 16], and thus that only some interventions should be allowed. While methodologically important, this is theoretically inessential. We can construe SCMs as possible representations of RCMs in the following sense:

**Definition 6.** Let $\mathcal{R} \in \mathfrak{R}_{\mathrm{eff}}$ and $\mathcal{M} \in \mathfrak{M}_{\mathrm{uniq}}$. We say that $\mathcal{M}$ *represents* $\mathcal{R}$ if its counterfactual distribution $P_{\mathrm{cf}}^\mathcal{M}$ marginalizes down to $P_{\mathrm{cf}}^\mathcal{R}$ on the potential outcomes defined by $\mathcal{R}$ (the set $\mathfrak{O}$ in Def. 1). We say $\mathcal{R}$ is *representable* if it is represented by at least some $\mathcal{M} \in \mathfrak{M}_{\mathrm{uniq}}$.[4]

Thus $\mathcal{M}$ represents $\mathcal{R}$ if they are counterfactually equivalent with respect to the outcomes defined by $\mathcal{R}$. Toward a characterization of representability, consider two properties of an RCM [9]:

**Definition 7.** The following Boolean formulas encode assumptions about potential outcomes:

$$\text{Composition.} \qquad Y_\mathbf{w}(u) = y \wedge Z_\mathbf{w}(u) = z \to Z_{\mathbf{w}y}(u) = z \qquad (2)$$

$$\text{Reversibility.} \qquad Y_{\mathbf{w}z}(u) = y \wedge Z_{\mathbf{w}y}(u) = z \to Y_\mathbf{w}(u) = y. \qquad (3)$$

Say $\mathcal{R} \in \mathfrak{R}_{\mathrm{eff}}$ satisfies composition and reversibility, respectively, when the respective statements hold for every unit $u$ of $\mathcal{R}$, whenever all the appropriate potential outcomes are defined.

We understand lower-case values like $y$, $z$, $\mathbf{w}$, when not bound as dummy indices or otherwise, to be schematic variables carrying tacit universal quantifiers. Thus (2), (3) must hold for all possible $y \in \mathrm{Val}(Y)$, $\mathbf{w} \in \mathrm{Val}(\mathbf{W})$, $z \in \mathrm{Val}(Z)$. This same usage is repeated, e.g., in (9).

While reversibility seems not to have arisen in the potential outcomes literature, instances of composition have appeared explicitly (e.g., Holland 12, p. 968) and have been used implicitly in concrete

---

[4]With regard to representability, our assumption that $\mathcal{R} \in \mathfrak{R}_{\mathrm{eff}}$ is without loss since Def. 4 easily implies that the potential outcomes induced by any SCM must satisfy effectiveness (1).

derivations (see Ex. 3 below). Note that the well-known principle of *consistency* [7, 27] is merely the instance of composition for $\mathbf{W} = \varnothing$. For $\mathcal{R}, \mathcal{R}' \in \mathfrak{R}_{\text{eff}}$ that share the same units $\mathcal{U}$ and endogenous variables $\mathbf{V}$ but have respective potential outcome sets $\mathfrak{O}, \mathfrak{O}'$ and potential response sets $\tilde{\mathfrak{F}}, \tilde{\mathfrak{F}}'$, if $\mathfrak{O} \subset \mathfrak{O}'$ and $\tilde{\mathfrak{F}} \subset \tilde{\mathfrak{F}}'$ we say that $\mathcal{R}'$ *extends* or is an extension of $\mathcal{R}$ and $\mathcal{R}$ is a *submodel* of $\mathcal{R}'$. Call $\mathcal{R}$ *full* if it has no proper extension. Then:

**Proposition 1** (SCM Representation). RCM $\mathcal{R}$ is representable iff $\mathcal{R}$ extends to some full $\mathcal{R}'$ that satisfies composition and reversibility.

Note that for an RCM $\mathcal{R}$ to be representable it is necessary (though not sufficient, in light of the models presented in Fig. 1 below) that $\mathcal{R}$ itself witness no composition or reversibility failures. Prop. 1 thus clarifies a sense in which RCMs are more general than SCMs, not just by allowing only a subset of allowable interventions, but also by imposing fewer requirements on how potential outcomes relate to one another. However, assuming composition, reversibility, and fullness, the two define the same classes of counterfactual distributions, despite the superficial differences in their definitions. In that case the two are, e.g., equivalent for interpreting the probabilistic logical language of §2. We submit that some version of this result makes sense of statements in the literature, e.g., from Pearl [27], that the twain are essentially equivalent frameworks from a theoretical perspective.

## 1.2 Causal Abstraction

The goal of this section is to clarify the source of putative failures of principles like composition. We suggest that it is helpful to view these issues through the lens of *causal abstraction* (the definitions in this section are adapted from [33, 5]). Abstraction has mostly been studied in the context of SCMs; our definitions apply equally to SCMs and RCMs via counterfactual distributions.

In causal abstraction, one has a set $\mathbf{V}_{\text{L}}$ of low-level (or concrete, or micro-) variables representing a fine-grained description and a set $\mathbf{V}_{\text{H}}$ of high-level (or abstract, or macro-) variables representing a coarser-grained description of the same scenario. The correspondence between the two descriptions is given by a partial *translation* map $\tau : \text{Val}(\mathbf{V}_{\text{L}}) \to \text{Val}(\mathbf{V}_{\text{H}})$. Translations extend canonically to maps of partial valuations (e.g., interventions) $\tau : \bigcup_{\mathbf{X} \subset \mathbf{V}_{\text{L}}} \text{Val}(\mathbf{X}) \to \bigcup_{\mathbf{X} \subset \mathbf{V}_{\text{H}}} \text{Val}(\mathbf{X})$ by setting $\tau(\mathbf{x}_{\text{L}}) = \mathbf{x}_{\text{H}}$ iff $\tau\big(\pi_{\mathbf{X}_{\text{L}}}^{-1}(\mathbf{x}_{\text{L}})\big) = \pi_{\mathbf{X}_{\text{H}}}^{-1}(\mathbf{x}_{\text{H}})$.

We overload $\tau$ once more so as to cover counterfactuals, defining as follows yet another partial $\tau : \text{Val}(\mathfrak{O}_{\text{L}}) \to \text{Val}(\mathfrak{O}_{\text{H}})$ for any sets $\mathfrak{O}_{\text{L}}, \mathfrak{O}_{\text{H}}$ of potential outcomes over $\mathbf{V}_{\text{L}}$ and $\mathbf{V}_{\text{H}}$ respectively. Index an element of $\text{Val}(\mathfrak{O}_{\text{L}})$ as $\{(\mathbf{y}_{\text{L}}^i)_{\mathbf{x}_{\text{L}}^i}\}_{1 \leq i \leq m}$, where $\mathbf{x}_{\text{L}}^i \neq \mathbf{x}_{\text{L}}^j$ for any $i \neq j$ and $\mathbf{y}_{\text{L}}^i \in \text{Val}(\{Y \in \mathbf{V}_{\text{L}} : Y_{\mathbf{x}_{\text{L}}^i} \in \mathfrak{O}_{\text{L}}\})$ for each $i$, and an element of $\text{Val}(\mathfrak{O}_{\text{H}})$ likewise as $\{(\mathbf{y}_{\text{H}}^j)_{\mathbf{x}_{\text{H}}^j}\}_{1 \leq j \leq n}$. Define $\tau\big(\{(\mathbf{y}_{\text{L}}^i)_{\mathbf{x}_{\text{L}}^i}\}_{1 \leq i \leq m}\big) = \{(\mathbf{y}_{\text{H}}^j)_{\mathbf{x}_{\text{H}}^j}\}_{1 \leq j \leq n}$ if $\tau(\{\mathbf{x}_{\text{L}}^i : 1 \leq i \leq m\}) = \{\mathbf{x}_{\text{H}}^j : 1 \leq j \leq n\}$ and $\tau(\mathbf{y}_{\text{L}}^i) = \mathbf{y}_{\text{H}}^j$ for any pair $\mathbf{x}_{\text{L}}^i, \mathbf{x}_{\text{H}}^j$ where $\tau(\mathbf{x}_{\text{L}}^i) = \mathbf{x}_{\text{H}}^j$.

**Definition 8.** With counterfactual translation in hand, we define an abstraction relation between probabilistic causal models. The model $\mathcal{H}$ abstracts $\mathcal{L}$ over the aligned variables (written $\mathcal{H} \prec_\tau \mathcal{L}$) if the translation $\tau$ pushes the latter's counterfactual distribution to the former's, that is, $P_{\text{cf}}^{\mathcal{H}} = \tau_*(P_{\text{cf}}^{\mathcal{L}})$.

A stricter and typically more useful notion is that of *constructive* abstraction (e.g., [5]). These arise from translations that can be generated variable-wise, and thus correspond to a coherent "clustering" of variables:

**Definition 9.** Translation $\tau : \text{Val}(\mathbf{V}_{\text{L}}) \to \text{Val}(\mathbf{V}_{\text{H}})$ is constructive if there is a partition $\Pi$ of $\mathbf{V}_{\text{L}}$ with non-overlapping cells $\{\Pi_V\}_{V \in \mathbf{V}_{\text{H}} \cup \{\perp\}}$, each $\Pi_V \subset \mathbf{V}_{\text{L}}$, where $\Pi_V$ is non-empty for all $V \neq \perp$, and a collection $\{\tau_V\}_{V \in \mathbf{V}_{\text{H}}}$ each of which is a partial surjective map $\tau_V : \text{Val}(\Pi_V) \to \text{Val}(V)$, such that $\tau(\mathbf{v}_{\text{L}}) = \big\{\tau_V\big(\pi_{\Pi_V}(\mathbf{v}_{\text{L}})\big)\big\}_{V \in \mathbf{V}_{\text{H}}}$ for any $\mathbf{v}_{\text{L}} \in \text{Val}(\mathbf{V}_{\text{L}})$.

A simple abstraction, ubiquitous in the literature (see, e.g., [16, §1.6.2] and [7]), is that of variable treatment levels. Here a higher-level value corresponds to some collection of lower-level specifications that might represent the potency or dosage of the drug administered, the time of administration, etc.: for example, a distinction of whether one took 300, 400, 500, or 600 mg of aspirin is made at the low level, but at the high level, there is only the binary distinction between having taken aspirin and not. Formally, a treatment variable $T$ is only binary with values $\text{c}, \text{tr}$ (control, treatment resp.) at the high level but takes on many values $\text{c}, \text{tr}^1, \dots, \text{tr}^n$ at the low level. The abstraction is made by omitting the fine-grained details; symbolically, one forms a new model by eliding the superscripts, collapsing

all $\mathrm{tr}^i$ into $\mathrm{tr}$. So long as for any outcomes we have $Y_{\mathrm{tr}^i}(u) = Y_{\mathrm{tr}^j}(u)$, the model thus formed will be a constructive probabilistic abstraction of the low-level model.

The next result provides some useful properties of constructive abstraction.

**Proposition 2.** Suppose $\mathcal{H} \prec_\tau \mathcal{L}$ with $\tau$ constructive. Then $\mathcal{H}$ is effective if $\mathcal{L}$ is effective. Also, for any submodel $\mathcal{H}'$ of $\mathcal{H}$ there is a submodel $\mathcal{L}'$ of $\mathcal{L}$ such that $\mathcal{H}' \prec_\tau \mathcal{L}'$.

Thus our general class of effective RCMs closes under constructive translation. The next example shows that this is not the case for the narrower class of representable models.

**Example 1.** Let $X, Y, X', Y'$ be variables with $\mathrm{Val}(X) = \{0, 1, 2\}$ and $\mathrm{Val}(X') = \mathrm{Val}(Y') = \mathrm{Val}(Y) = \{0, 1\}$. Consider the RCM $\mathcal{R}_\mathrm{L}$ defined over $\mathbf{V}_\mathrm{L} = \{X, Y\}$ as a conjunction of POs:

$$X = 1 \wedge Y = 1 \wedge Y_{X=2} = 0 \wedge X_{Y=0} = 1 \tag{4}$$

for a single unit (suppressed above for clarity). Consider a second RCM $\mathcal{R}_\mathrm{H}$ over $\mathbf{V}_\mathrm{H} = \{X', Y'\}$:

$$X' = 1 \wedge Y' = 1 \wedge Y'_{X'=1} = 0 \wedge X'_{Y'=0} = 1. \tag{5}$$

Note that $\mathcal{R}_\mathrm{H} \prec_\tau \mathcal{R}_\mathrm{L}$ where $\tau$ is a constructive abstraction with $\Pi_{X'} = \{X\}$, $\Pi_{Y'} = \{Y\}$ and $\tau(X = 0) = 0$, $\tau(X = 1) = \tau(X = 2) = 1$, $\tau(Y = y) = y$. Now $\mathcal{R}_\mathrm{H}$ violates both composition and reversibility, while $\mathcal{R}_\mathrm{L}$ is representable.

A second observation is that the analogue of the claim about submodels in Prop. 2 does not hold for extensions:

**Example 2.** Consider enlarging (4) with the potential outcome $Y_{X=1} = 1$. Then there is no high-level abstraction under $\tau$ that defines the outcome $Y'_{X'=1}$, since $Y_{X=2} = 0 \wedge Y_{X=1} = 1$ translates to $Y'_{X'=1} = 0 \wedge Y'_{X'=1} = 1$.

The main result of this section is that the phenomenon exhibited by Ex. 1 accounts for all representability failures:

**Theorem 1** (Abstract Representation). Let $\mathcal{R}$ be an RCM. Then there is a representable $\mathcal{R}_\mathrm{L}$ and constructive translation $\tau$ such that $\mathcal{R} \prec_\tau \mathcal{R}_\mathrm{L}$.

It is worth remarking on the connection between a well-known twofold condition called the Stable Unit Treatment Value Assumption (SUTVA [16, §1.6]) and causal abstraction. The first part of SUTVA is the assumption that "the potential outcomes for any unit do not vary with the treatments assigned to other units"; this is already presumed by our definition of causal model, which does not admit interventions on multiple units (however, see Rmk. A.1 for a way to model this condition within our framework as an application of abstraction). The second part is that "for each unit, there are no different forms or versions of each treatment level, which lead to different potential outcomes." Note that this assumption can be seen as guaranteeing the viability of the variable treatment levels abstraction, as it is simply a restatement of the condition we already identified—that the outcomes $Y_{\mathrm{tr}^i}(u)$ for any unit $u$ and treatment level $\mathrm{tr}^i$ must all agree.[5]

## 2 Inference

The raison d'être of a causal inference framework is to provide a language for encoding causal assumptions and showing when and why conclusions follow from available data and appropriate assumptions. In this section, to provide further granularity on the comparison between RCMs and SCMs, we introduce a neutral formal language that is naturally interpreted relative to both of these models. The language systematizes reasoning about the probabilities of counterfactuals. Fixing a set $\mathfrak{O}$ of potential outcome pairs, we define a formal language $\mathcal{L}$ in two stages:

**Definition 10.** The *base language* $\mathcal{L}_{\mathrm{base}}$ is given by all Boolean combinations of statements $Y_\mathbf{x} = y$, alternatively written $y_\mathbf{x}$, for all $Y_\mathbf{x} \in \mathfrak{O}$, $y \in \mathrm{Val}(Y)$. Meanwhile, $\mathcal{L}$ is defined as the set of Boolean combinations of inequalities $\mathbf{t}_1 \geqslant \mathbf{t}_2$, where $\mathbf{t}_1, \mathbf{t}_2$ are generated as sums, products, and additive inverses of probability terms $\mathbf{P}(\varepsilon)$, where $\varepsilon \in \mathcal{L}_{\mathrm{base}}$.

---

[5]Imbens and Rubin [16] also mention ways of fulfilling this condition requiring changes to the causal model definition. In the supplement (Rmk. A.2) we show these alternative models can be represented within our framework.

The language $\mathcal{L}$ is the most expressive in a sequence of three languages introduced in [13, 4] to formalize the "causal hierarchy" [27]. By restricting probability terms to purely "observational" or "interventional" quantities, it is possible to study the inferential limitations of data and assumptions at lower levels of this hierarchy. For present purposes, $\mathcal{L}$ naturally encodes prominent reasoning patterns in RCMs and in SCMs. Its semantics are straightforward in any $\mathcal{M}$ or $\mathcal{R}$ that includes all outcomes $\mathfrak{O}$: we generate a mapping of each polynomial term $\mathbf{t} \mapsto [\![\mathbf{t}]\!] \in \mathbb{R}$ recursively with the crucial case being to map $\mathbf{P}(\varepsilon)$ to the probability calculable by marginalization of $p_{\mathrm{cf}}^{\mathcal{R}}$ or $p_{\mathrm{cf}}^{\mathcal{M}}$, and then evaluate the atom $\mathbf{t}_1 \geqslant \mathbf{t}_2$ true iff $[\![\mathbf{t}_1]\!] \geq [\![\mathbf{t}_2]\!]$, recursing to define a semantics for all of $\mathcal{L}$. Over the class of all (recursive, possibly infinite) SCMs, $\mathcal{L}$ has been axiomatized [13] by a set of principles called $\mathsf{AX}_3$, and the complexity of its satisfiability problem has been shown complete for the class $\exists\mathbb{R}$ [24]. The class of simple probability distributions over the atoms of $\mathcal{L}_{\mathrm{base}}$ is axiomatized by principles known as $\mathsf{AX}_1$ [13], which we will abbreviate $\mathsf{AX}$.

## 2.1 Potential Outcomes Assumptions

Reasoning about potential outcomes is often encoded in what we call the base language $\mathcal{L}_{\mathrm{base}}$, augmented with (typically implicit universal) quantifiers over units. For instance, the well known *monotonicity* (or "no defiers" who do the opposite of their prescription) assumption [15, 16] says

$$\forall u.X_{z^-}(u) = 1 \rightarrow X_{z^+}(u) = 1, \tag{6}$$

where $X$ and $Z$ are binary variables respectively meaning the treatment (actually taken) and the treatment prescribed, with $z^+, z^-$ abbreviating $Z = 1, Z = 0$ respectively. We will use the same abbreviation for other binary variables, so that the above condition can be written succinctly as $x_{z^+}^+ \rightarrow x_{z^-}^+$. We also adopt this interpretation of $X, Z$ for the rest of §2.1. We now explain how this and other causal assumptions in the potential outcomes framework can be encoded in $\mathcal{L}$:

**Definition 11.** Let $\xi$ be a well-formed, closed predicate formula in prenex normal form with a single quantifier over a variable $\{u\}$ and matrix in $\mathcal{L}_{\mathrm{base}}$; the $u$ can alternately be included in the atoms, e.g., by writing $Y_{\mathbf{x}}(u) = y$. Define its encoding $\mathrm{T}(\xi) \in \mathcal{L}$ as follows:

$$\mathrm{T}(\xi) = \begin{cases} \mathbf{P}(\neg\mathrm{T}(\zeta)) = 0, & \xi = \forall u.\zeta, \text{ where } \zeta \in \mathcal{L}_{\mathrm{base}} \\ \mathbf{P}(\mathrm{T}(\zeta)) > 0, & \xi = \exists u.\zeta, \text{ where } \zeta \in \mathcal{L}_{\mathrm{base}} \end{cases}.$$

Note that $\zeta$ is quantifier-free in both cases. Thus, e.g., the encoding of (6) is $\mathbf{P}\big[\neg(x_{z^-}^+ \rightarrow x_{z^+}^+)\big] = 0$.

Where $S$ is a set of $\mathcal{L}_{\mathrm{base}}$ assumptions let $\mathfrak{R}(S)$ be the class of RCMs whose potential outcomes obey every assumption in $S$, thus obeying $\forall u.\sigma$ where $u$ ranges over units for any $\sigma \in S$. Also let $\mathrm{T}(S) = \{\mathrm{T}(\forall u.\sigma)\}_{\sigma \in S}$ be the encoding of $S$ via Def. 11. Then we have the following:

**Theorem 2.** $\mathsf{AX} + \mathrm{T}(S)$ is sound and complete for $\mathfrak{R}(S)$.

**Corollary 1.** $\mathsf{RCM} = \mathsf{AX} + \mathrm{T}(1)$ is sound and complete for $\mathfrak{R}_{\mathrm{eff}}$.

One consequence of this completeness result is that purely propositional and predicate logic reasoning about potential outcomes can be interweaved with probabilistic reasoning, as in Ex. 3 below. Another consequence is a complete axiomatization of SCMs (which can be seen as a probabilistic lift of [10]):

**Corollary 2.** Let $\mathsf{C}, \mathsf{Rev}$ be universal statements of (2) and (3) respectively. Then $\mathsf{SCM} = \mathsf{RCM} + \mathrm{T}(\mathsf{C}) + \mathrm{T}(\mathsf{Rev})$ is sound and complete for $\mathfrak{M}_{\mathrm{uniq}}$ (where every outcome is included in $\mathcal{L}_{\mathrm{base}}$).

**Example 3.** A seminal result from [15, 1] is that it is possible to estimate the Average Treatment Effect among the population of units who *comply* with their treatment assignment, a quantity known as the *Local* Average Treatment Effect (LATE): $\mathbf{E}(Y_{x^+} - Y_{x^-} | x_{z^+}^+ \wedge x_{z^-}^-)$, with $Y$ the outcome, which we assume binary purely for simplicity, and without loss of generality. Thm. 2 implies that this can be verified in our calculus, by appeal to two key assumptions [15, 1]: monotonicity (6) and

$$\text{Exclusion restriction (ER).} \qquad \forall u.y_{z^-,x}(u) \leftrightarrow y_{z^+,x}(u). \tag{7}$$

The original discovery was that these principles guarantee that $\mathrm{LATE} = \mathrm{ITT}_1/\mathrm{ITT}_2$, where the latter are the average "causal effects of assignment on treatment received and on the outcome of interest" [1], or symbolically:

$$\mathrm{ITT}_2 = \mathbf{E}(X_{z^+} - X_{z^-}) = \mathbf{P}(x_{z^+}^+ \wedge x_{z^-}^-) - \mathbf{P}(x_{z^+}^- \wedge x_{z^-}^+)$$

$$\mathrm{ITT}_1 = \mathbf{E}\big(Y_{z^+,X_{z^+}} - Y_{z^-,X_{z^-}}\big) = \mathbf{P}\big(y_{z^+,X_{z^+}}^+ \wedge y_{z^-,X_{z^-}}^-\big) - \mathbf{P}\big(y_{z^+,X_{z^+}}^- \wedge y_{z^-,X_{z^-}}^+\big).$$

| | | | | $\mathcal{R}(\varepsilon)$ | | | |
|---|---|---|---|---|---|---|---|
| $u$ | $P(u)$ | $X_{z^+}(u)$ | $X_{z^-}(u)$ | $Y_{x^+,z}(u)$ | $Y_{x^-,z}(u)$ | $Y_{x^+}(u)$ | $Y_{x^-}(u)$ |
| $u_0$ | $3/4 - \varepsilon$ | 1 | 0 | 1 | 0 | 1 | 0 |
| $u_1$ | $1/4 - \varepsilon$ | 1 | 0 | 0 | 1 | 0 | 1 |
| $u_2$ | $\varepsilon$ | 1 | 0 | 0 | 0 | 1 | 0 |
| $u_3$ | $\varepsilon$ | 1 | 0 | 1 | 1 | 1 | 1 |

Figure 1: A family of RCMs $\{\mathcal{R}(\varepsilon)\}_{0 \leq \varepsilon \leq 1/4}$ such that $\mathcal{R}(\varepsilon)$ is representable iff $\varepsilon = 0$ (though (2) and (3) are met for any unit). Note that all members of this family meet (6) and (7) (the latter guaranteed since columns $Y_{x^+,z}(u)$, $Y_{x^-,z}(u)$ give the potential outcome for any $z$). Also, any experimentally testable quantities—including, in particular, $\text{ITT}_1$ and $\text{ITT}_2$—agree across the family, with $\frac{\text{ITT}_1}{\text{ITT}_2} = 1/2$. However the assumption in question (8) holds in $\mathcal{R}(\varepsilon)$ iff $\varepsilon = 0$, and LATE $= 1/2 + \varepsilon$, so that LATE $= \frac{\text{ITT}_1}{\text{ITT}_2}$ only when this holds, and it is not estimable in general.

where in $\text{ITT}_1$, interventions like $X_z$ set $X$ at the unit level to the value that it would take under the intervention setting $Z$ to $z$; thus, e.g., we have $\mathbf{P}(y_{z,X_z}) = \mathbf{P}(y_{z,x^+} \wedge x_z^+) + \mathbf{P}(y_{z,x^-} \wedge x_z^-)$. Crucially, these two quantities can be estimated, e.g., through randomized experiments [16, Ch. 23].

However, over our most general class $\mathfrak{R}_{\text{eff}}$ of RCMs, these two assumptions are not in fact sufficient to identify LATE. Fig. 1 illustrates a family of RCMs that satisfy (6) and (7), but disagree on LATE. An additional principle, which Angrist et al. [1] offer as a matter of notation, we dub:

$$\text{Outcome decomposition.} \qquad \forall u. y_x(u) \leftrightarrow y_{z^+,x}(u). \qquad (8)$$

It can then be shown that, taken together, (6), (7), and (8) do indeed logically entail LATE $= \text{ITT}_1/\text{ITT}_2$; see Prop. B.1 in the technical appendix for the derivation.

There has been much discussion of monotonicity and exclusion restrictions (which are closely related to graphical assumptions; see §2.2 below), but what might justify outcome decomposition (8)? One intuition might be that it somehow follows from the exclusion restriction (7): if the effect of $X$ on $Y$ is the same no matter the value of $Z$, then it would seem that omitting $z^+$ in the intervention should have no impact on the effect of $X$ on $Y$. Of course, the example in Fig. 1 shows that this is too quick.

It turns out that (8) does follow from (7) if we restrict attention to *representable* RCMs. In fact, (8) is derivable from (7) and the principle of *composition* (2) in the calculus AX, so long as we can reason along the way about the potential response $Z_x$. By composition, for any $x$ and $y$ we have $y_x \wedge z_x^+ \rightarrow y_{z^+,x}$ and $y_x \wedge z_x^- \rightarrow y_{z^-,x}$, and by ER (7) the latter gives $y_x \wedge z_x^- \rightarrow y_{z^+,x}$. As $Z_x$ is binary, we have $z_x^+ \vee z_x^-$, and thus by propositional reasoning, $y_x \rightarrow y_{z^+,x}$. The other direction $y_{z^+,x} \rightarrow y_x$ follows from the same argument by contraposition, as $Y$ too is binary.

Thus, while the full power of composition is not invoked, it is natural to read this example and much of the literature as implicitly assuming something like representability (thus implying composition). Another source of support for this is that under representability one can show (see Prop. B.2) that $\text{ITT}_1 = \mathbf{E}(Y_{z^+} - Y_{z^-})$, an even simpler and manifestly identifiable expression for this average effect.

## 2.2 Graphical Assumptions

As we saw above (Prop. 1), SCMs can be understood as *representations* of suitable RCMs. As such, they also suggest further sources of assumptions for deriving causal inferences. In particular, qualitative patterns of functional dependence introduce the possibility of *graphical* methods:

**Definition 12.** Let $\mathcal{M} = \langle \mathbf{U}, \mathbf{V}, \{f_V\}_{V \in \mathbf{V}}, P \rangle$ be an SCM. Then define the *causal diagram* $\mathcal{G}(\mathcal{M})$ of $\mathcal{M}$ as a graph over nodes $\mathbf{V}$, with mixed directed edges $\rightarrow$ and *bidirected arcs* $\leftarrow\!-\!-\!\rightarrow$. For any $V, V' \in \mathbf{V}$, there is a directed edge $V \rightarrow V'$ if $V \in \mathbf{Pa}_{V'}$, and there is a bidirected edge $V \leftarrow\!-\!-\!-\!\rightarrow V'$ if $\mathbf{U}_V, \mathbf{U}_{V'} \subset \mathbf{U}$ are correlated under $P$ (including if $\mathbf{U}_V \cap \mathbf{U}_{V'} \neq \varnothing$).

Letting $\mathfrak{M}(\mathcal{G})$ be the set of SCMs with diagram $\mathcal{G}$, we have $\mathfrak{M}(\mathcal{G}) \subset \mathfrak{M}_{\text{uniq}}$ provided the directed edges in $\mathcal{G}$ form a dag (see Lem. A.3). We thus assume this acyclicity of any $\mathcal{G}$. When interpreting over an SCM, we include every possible potential outcome in $\mathcal{L}$. Just as we earlier encoded assumptions about the potential outcomes of an RCM into $\mathcal{L}$, we may do the same for SCMs regarding their graphs. A first observation is that Def. 11 translates axiom C6 of [10] to ProbRec of [13], thus

rederiving the system $\mathsf{AX}_3$ for the class of all acyclic SCMs, i.e. $\bigcup_{\mathcal{G}} \mathfrak{M}(\mathcal{G})$, from the latter. We now encode the content of (the assumption of having) a *particular* diagram $\mathcal{G}$ into $\mathcal{L}$. Let $\mathbf{Pa}_V^{\mathcal{G}} = \{V' \in \mathbf{V} : V' \to V \in \mathcal{G}\}$ be the directed parents in a graph $\mathcal{G}$ of a vertex $V$. We encode by way of two schemas, encapsulating what some [28] have called "the two fundamental laws of causal inference":

**Definition 13.** Let the exclusion restriction schema $\mathsf{ER}^{\mathcal{G}}$ be the $\mathcal{L}_{\text{base}}$ principle $y_{\mathbf{a}} \leftrightarrow y_{\mathbf{p}}$, for all variables $Y \in \mathbf{V}$ and sets of variables $\mathbf{A} \supset \mathbf{Pa}_V^{\mathcal{G}}$, where $y \in \text{Val}(Y)$, $\mathbf{a} \in \text{Val}(\mathbf{A})$, $\mathbf{p} = \pi_{\mathbf{Pa}_V^{\mathcal{G}}}(\mathbf{a})$. Let the counterfactual independence schema $\mathsf{cf\text{-}sep}^{\mathcal{G}}$ be, for all pairs of variable sets $\{Y_i\}_{1 \leq i \leq n}, \{Y_j'\}_{1 \leq j \leq n'} \subset \mathbf{V}$ such that there are no $Y_i, Y_j'$ for which $Y_i = Y_j'$ or $Y_i \leftarrow\!\!-\!\!-\!\!-\!\!\to Y_j'$ in $\mathcal{G}$,

$$\mathsf{cf\text{-}sep}^{\mathcal{G}}. \qquad \mathbf{P}\Big[ \bigwedge_{1 \leq i \leq n} (y_i)_{\mathbf{p}_i} \wedge \bigwedge_{1 \leq j \leq n'} (y_j')_{\mathbf{p}_j'} \Big] = \mathbf{P}\Big[ \bigwedge_{1 \leq i \leq n} (y_i)_{\mathbf{p}_i} \Big] \cdot \mathbf{P}\Big[ \bigwedge_{1 \leq j \leq n'} (y_j')_{\mathbf{p}_j} \Big] \quad (9)$$

where $y_i \in \text{Val}(Y_i)$, $y_j' \in \text{Val}(Y_j')$, $\mathbf{p}_i \in \text{Val}(\mathbf{Pa}_{Y_i}^{\mathcal{G}})$, $\mathbf{p}_j' \in \text{Val}(\mathbf{Pa}_{Y_j'}^{\mathcal{G}})$ for each $Y_i$, $Y_j'$. Then the translation of $\mathcal{G}$ is the combination of axioms $\mathrm{T}(\mathcal{G}) = \mathrm{T}(\mathsf{ER}^{\mathcal{G}}) + \mathsf{cf\text{-}sep}^{\mathcal{G}}$.

Note that while Ex. 3 in no way relies on graphs, if we accept a $\mathcal{G}$ where $Z \not\to Y$, then $\mathsf{ER}^{\mathcal{G}}$ yields $y_x \leftrightarrow y_{zx} \leftrightarrow y_{z'x}$ without further ado. Importantly, however, T(6) is not valid over $\mathfrak{M}(\mathcal{G})$ for any $\mathcal{G}$ containing the edge $Z \to X$, revealing an extra-graphical provenance. On the other hand, $\mathsf{cf\text{-}sep}$ is inexpressible in $\mathcal{L}_{\text{base}}$—inferentially, the two approaches are incomparable.

A long-standing question has been whether exclusion restriction and independence axioms together could be *complete*, in that they capture all the inferential content of a given causal diagram $\mathcal{G}$ (see, e.g., [39, 8, 31]). Answering such questions can help with the development of tractable inference methods. Partial completeness results for limited queries are known [35], and the method from Tian [39] supplies an algorithm that is complete with respect to all equality constraints [8]. Placing no limitations on queries beyond their expressibility in $\mathcal{L}$—and thus including inequality constraints as well—but making certain restrictions on $\mathcal{G}$, we answer this question in the affirmative:

**Theorem 3.** For any acyclic diagram $\mathcal{G}$, axioms $\mathrm{T}(\mathcal{G}) + \mathsf{SCM}$ are sound for $\mathcal{L}$ over $\mathfrak{M}(\mathcal{G})$, and also complete if the bidirected arcs in $\mathcal{G}$ form a disjoint union of complete graphs.

Often the famous *d-separation* conditional independence criterion (Def. B.2) is used in place of our $\mathsf{cf\text{-}sep}$. Since all instances of the latter are instances of the former, our Thm. 3 is stronger (see Cor. B.1). This completeness result implies that for such a $\mathcal{G}$, any known graphical conclusions—including *do*-calculus, identifiability results, and bounds—can be rederived in our calculus, e.g.:

**Example 4** (Verma constraints)**.** We derive the *Verma constraint* [40, 39] over the graph $\mathcal{G}$ of Fig. 2a that $\sum_w \mathbf{P}(y \mid z, w, x)\mathbf{P}(w \mid x)$ does not depend functionally on $x$:

$$\sum_w \frac{\mathbf{P}(y, z, w, x)\mathbf{P}(w, x)}{\mathbf{P}(z, w, x)\mathbf{P}(x)} \stackrel{\mathrm{T(C)}}{=} \sum_w \frac{\mathbf{P}(y_{zwx}, z_{ywx}, w_{yzx}, x_{yzw})\mathbf{P}(w_x, x_w)}{\mathbf{P}(z_{wx}, w_{zx}, x_{zw})\mathbf{P}(x)}$$

$$\stackrel{\mathrm{T(ER}^{\mathcal{G}})}{=} \sum_w \frac{\mathbf{P}(y_z, z_w, w_x, x)\mathbf{P}(w_x, x)}{\mathbf{P}(z_w, w_x, x)\mathbf{P}(x)} \stackrel{(9)}{=} \sum_w \frac{\mathbf{P}(y_z, w_x)\mathbf{P}(z_w)\cancel{\mathbf{P}(x)}\cancel{\mathbf{P}(w_x)}\cancel{\mathbf{P}(x)}}{\cancel{\mathbf{P}(z_w)}\cancel{\mathbf{P}(w_x)}\cancel{\mathbf{P}(x)}\mathbf{P}(x)} = \mathbf{P}(y_z).$$

To summarize, for a wide class of cases, Thm. 3 tells us that the two principles encoded in $\mathrm{T}(\mathcal{G})$ exhaust the types of algebraic statements that a researcher is committed to when venturing a graphical assumption. Putting this in algebraic terms facilitates a perspective on such assumptions that can be naturally interpreted with respect to RCMs, independent of any representation by an SCM.

## 2.3 Single-World Intervention Graphs

Another graphical framework that draws on ideas and concepts from both frameworks is that of *single-world intervention graphs* (SWIGs) [30]. In comparison to the usual formulation of do-calculus, SWIGs facilitate reasoning with a wider class of expressions by combining graphical and potential outcome notations (see, e.g., [20]). Here we show that this "hybrid" framework can also naturally be assimilated to the logical perspective adopted in the present paper. Assuming acyclicity:

**Definition 14.** Let $\mathcal{D}$ be a dag over $\mathbf{V}$ and let $\mathbf{x}$ be an intervention. Let $\mathbf{An}_V^{\mathcal{D}} = \{V' \in \mathbf{V} : V' \to \cdots \to V \text{ in } \mathcal{D}\}$ be the directed ancestors[6] of a variable $V$ in $\mathcal{D}$. Then the SWIG $\mathcal{D}_{\mathbf{x}}$ has nodes labeled

---

[6]I.e., the transitive closure of the converse of the parent sets $\mathbf{Pa}_V^{\mathcal{D}}$. Note that $V \in \mathbf{An}_V^{\mathcal{D}}$ and $\mathbf{Pa}_V^{\mathcal{D}} \subset \mathbf{An}_V^{\mathcal{D}}$.

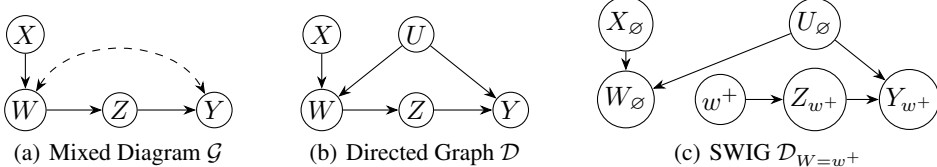

(a) Mixed Diagram $\mathcal{G}$      (b) Directed Graph $\mathcal{D}$      (c) SWIG $\mathcal{D}_{W=w^+}$

Figure 2: (a) Graph $\mathcal{G}$ for Ex. 4. Note that we have $\mathsf{ER}^{\mathcal{G}} \vDash y_{zwx} \leftrightarrow y_z, z_{ywx} \leftrightarrow z_w, w_{yzx} \leftrightarrow w_x,$
$x_{yzw} \leftrightarrow x, x_w \leftrightarrow x, z_{wx} \leftrightarrow z_w,$ etc., while pairs where $\mathsf{cf\text{-}sep}^{\mathcal{G}}$ is applicable include $\{W, Y\},$
$\{X, Z\}$ and $\{Z\}, \{W, X\}$ (and any pairwise subsets thereof). These instances are used in Ex. 4.
(b) Directed graph $\mathcal{D}$, obtained from $\mathcal{G}$ by making the latent exogenous variable $U$ explicit.
(c) SWIG obtained from $\mathcal{D}$ for the intervention $W = w^+$, with fixed node $w^+$. (10) implies
$\mathbf{P}(w, y_{w^+}, z_{w^+})\mathbf{P}(u) = \mathbf{P}(w, u)\mathbf{P}(y_{w^+}, z_{w^+}, u)$ since every path between $W_{\varnothing}$ and either $Z_{w^+}$ or
$Y_{w^+}$ goes through $U_{\varnothing}$, where the directions of the arrows do not collide.

$\{V_{\pi_{\mathbf{A}}(\mathbf{x})} : V \in \mathbf{V}\} \cup \{\pi_V(\mathbf{x}) : V \in \mathbf{X}\}$, where $\mathbf{A} = (\mathbf{An}_V^{\mathcal{D}} \cap \mathbf{X}) \setminus \{V\}$, with nodes in the first set
in this union termed *random*, and those in the second termed *fixed*. This SWIG has edges

$$\{\pi_X(\mathbf{x}) \to V_{\mathbf{p}} : X \in \mathbf{X}, \mathcal{G} \text{ has edge } X \to V\} \cup \{V_{\mathbf{p}} \to V'_{\mathbf{p}'} : V \notin \mathbf{X}, \mathcal{G} \text{ has edge } V \to V'\}.$$

See Fig. 2c for an example of this construction. Note that edges in the first set in the union above
have fixed heads and random tails, while those in the second set have random heads and tails.

**Definition 15.** Define the following conditional independence schema $\mathsf{sw\text{-}sep}^{\mathcal{D}_{\mathbf{x}}}$, for any sets of
random nodes $\{(X_i)_{\mathbf{p}_i} : 1 \le i \le l\}$ and $\{(Y_j)_{\mathbf{p}'_j} : 1 \le j \le m\}$ that are d-separated (Def. B.1) given
$\{(Z_k)_{\mathbf{p}''_k} : 1 \le k \le n\}$ in the SWIG $\mathcal{D}_{\mathbf{x}}$:

$$\mathsf{sw\text{-}sep}^{\mathcal{D}_{\mathbf{x}}}. \quad \mathbf{P}\Big[ \bigwedge_{\substack{1 \le i \le l \\ 1 \le j \le m}} (x_i)_{\mathbf{p}_i} \wedge (y_j)_{\mathbf{p}'_j} \Big] \cdot \mathbf{P}\Big[ \bigwedge_{1 \le k \le n} (z_k)_{\mathbf{p}''_k} \Big]$$

$$= \mathbf{P}\Big[ \bigwedge_{\substack{1 \le i \le l \\ 1 \le k \le n}} (x_i)_{\mathbf{p}_i} \wedge (z_k)_{\mathbf{p}''_k} \Big] \cdot \mathbf{P}\Big[ \bigwedge_{\substack{1 \le j \le m \\ 1 \le k \le n}} (y_j)_{\mathbf{p}'_j} \wedge (z_k)_{\mathbf{p}''_k} \Big]. \quad (10)$$

One notable model associated with SWIGs is the *FFRCISTG* [32]; given the same graph, FFRCISTGs
are compatible with SCMs [30] but issue fewer (potentially controversial) implications:

**Definition 16.** Let $\mathcal{R}$ be a full RCM. Then $\mathcal{R}$ is a *FFRCISTG over* $\mathcal{D}$ if every instance of $\mathrm{T}(\mathsf{ER}^{\mathcal{D}})$
and $\mathsf{sw\text{-}sep}^{\mathcal{D}}$ holds in its counterfactual distribution. Let $\mathfrak{F}(\mathcal{D})$ be the class of FFRCISTGs over $\mathcal{D}$.

**Proposition 3.** Suppose the SCM $\mathcal{M} \in \mathfrak{M}(\mathcal{D})$ represents the full RCM $\mathcal{R}$. Then $\mathcal{R} \in \mathfrak{F}(\mathcal{D})$.   $\square$

Given that Def. 16 already defines $\mathfrak{F}(\mathcal{D})$ in terms of $\mathcal{L}$-principles, while [30] have shown the soundness
direction, the following is straightforward:

**Theorem 4.** $\mathsf{RCM} + \mathrm{T}(\mathsf{ER}^{\mathcal{D}}) + \bigcup_{\mathbf{x}} \mathsf{sw\text{-}sep}^{\mathcal{D}_{\mathbf{x}}}$ is sound and complete over $\mathfrak{F}(\mathcal{D})$.   $\square$

## 3 Conclusion

The task of this paper has been to clarify the senses in which the Rubin causal model and structural
causal models are very closely related formalisms for encoding causal assumptions and deriving
causal conclusions. We concur with [23], [14], [41] and others that "there are insights that arise when
using each that are less transparent when using the other" [41, p. 8]. Our interest in this paper has
been to elucidate the comparison from a theoretical ("in principle") perspective.

We do not suppose that the present work will be the final word on theoretical connections between
RCMs and SCMs. On the contrary, there remain numerous open questions. Perhaps chief among
these is the generalization of Thm. 3 to encompass all possible causal diagrams (not just those in
which the bidirected arcs form a disjoint union of complete graphs). Does the theorem hold with
no further principles, or do additional algebraic constraints arise? This important open question
[39, 8, 31] is a crucial step toward a complete theoretical synthesis of the two frameworks.

**Acknowledgments**

This work was partially supported by a seed grant from the Stanford Institute for Human-Centered Artificial Intelligence. We also thank Elias Bareinboim and Guido Imbens for helpful conversations.

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
