# Comparing Causal Frameworks: Potential Outcomes, Structural Models, Graphs, and Abstractions (Supplement)

These appendices contain demonstrations of the results in the main text as well as additional technical notes.

## A   Modeling (§1)

### A.1   Preliminaries

**Proposition A.1.** Uniform RCMs are dense (in the weak subspace topology on counterfactual distributions[1]): for any $\mathcal{R}$, $\varepsilon > 0$, there is a $\mathcal{R}'$ whose distribution is uniform that $\varepsilon$-approximates $\mathcal{R}$ (in, e.g., the standard metric on counterfactual distributions considered as points in Euclidean space, or the Lévy-Prohorov metric).

*Proof.* Follows by density of $\mathbb{Q} \subset \mathbb{R}$ since any RCM with a rational distribution is representable also by a uniform distribution on units. □

*Proof of Prop. 1.* Suppose $\mathcal{R} = \langle \mathcal{U}, \mathbf{V}, \mathfrak{D}, \{f_{Y_\mathbf{x}}\}_{Y_\mathbf{x} \in \mathfrak{D}}, P \rangle$ is representable. Then we have an SCM $\mathcal{M} = \langle \mathbf{U}, \mathbf{V}, \{f_V\}_V, P' \rangle \in \mathfrak{M}_{\text{uniq}}$ inducing its counterfactual distribution on $\text{Val}(\mathfrak{D})$. For each $u \in \mathcal{U}$, there must be a $\mathbf{u}_u \in \text{Val}(\mathbf{U})$ such that $P'(\mathbf{u}_u) > 0$ and $\mathcal{M}_\mathbf{x}, \mathbf{u}_u \vDash Y = f_{Y_\mathbf{x}}(u)$ for each $Y_\mathbf{x} \in \mathfrak{D}$. Consider the RCM $\mathcal{R}' = \langle \mathcal{U}, \mathbf{V}, \{Y_\mathbf{x}\}_{\text{all } Y_\mathbf{x}}, \{f'_{Y_\mathbf{x}}\}_{Y_\mathbf{x}}, P \rangle$ where we define $f'_{Y_\mathbf{x}}(u) = f_{Y_\mathbf{x}}(u)$ for each $Y_\mathbf{x} \in \mathfrak{D}$, and $f'_{Y_\mathbf{x}} = \pi_Y(\mathbf{v})$ for each $Y_\mathbf{x} \notin \mathfrak{D}$ where $\mathbf{v}$ is the unique solution such that $\mathcal{M}_\mathbf{x}, \mathbf{u}_u \vDash \mathbf{v}$. Note that $\mathcal{R}'$ extends $\mathcal{R}$ by construction, has no proper extension itself, and satisfies composition and reversibility by the soundness direction of [2, Thm. 3.2], as these principles must apply to $\mathcal{M}$.

Conversely suppose we have the extension $\mathcal{R}'$. Then by the completeness direction of the proof of [2, Thm. 3.2] we can construct an SCM $\mathcal{M} \in \mathfrak{M}_{\text{uniq}}$ representing $\mathcal{R}'$ and thus also $\mathcal{R}$. Specifically, we can derive a unique maximal consistent set $\Gamma$ from the outcomes defined in $\mathcal{R}'$ (unique maximality is guaranteed since $\mathcal{R}'$ has no proper extension and consistency since $\mathcal{R}'$ meets composition and reversibility), and from there define the equations of $\mathcal{M}$. □

### A.2   Causal Abstraction

In this section we will use the following technical result to decompose constructive partial translations.

**Lemma A.1.** Suppose $\tau$ is a constructive translation of $\mathbf{V}_\text{L}$ to $\mathbf{V}_\text{H}$ and let $\mathbf{x}_\text{L}, \mathbf{x}_\text{H}$ be partial settings of $\mathbf{X}_\text{L} \subset \mathbf{V}_\text{L}, \mathbf{X}_\text{H} \subset \mathbf{V}_\text{H}$ respectively. Then $\tau(\mathbf{x}_\text{L}) = \mathbf{x}_\text{H}$ iff:

1. For each $X_\text{H} \in \mathbf{X}_\text{H}$, we have that $\tau_{X_\text{H}}(\theta_{X_\text{H}}) = \pi_{X_\text{H}}(\mathbf{x}_\text{H})$ where we define $\theta_{V_\text{H}} = \{\mathbf{x} \in \text{Val}(\Pi_{V_\text{H}}) : \pi_{\mathbf{X}_\text{L} \cap \Pi_{V_\text{H}}}(\mathbf{x}) = \pi_{\mathbf{X}_\text{L} \cap \Pi_{V_\text{H}}}(\mathbf{x}_\text{L})\}$ for each $V_\text{H} \in \mathbf{V}_\text{H}$.

2. For each $V_\text{H} \notin \mathbf{X}_\text{H}$, we have that $\tau_{V_\text{H}}(\theta_{V_\text{H}}) = \text{Val}(V_\text{H})$.

---

[1] See [4, 1] for more detail on this topology and inducing metrics.

37th Conference on Neural Information Processing Systems (NeurIPS 2023).

*Proof.* First we show the "if" direction. It follows easily from 1 that $\tau(\mathbf{v}_L) \in \pi_{\mathbf{X}_H}^{-1}(\mathbf{x}_H)$ for any $\mathbf{v}_L \in \pi_{\mathbf{X}_L}^{-1}(\mathbf{x}_L)$, and it follows from 2 and 1 that for any $\mathbf{v}_H \in \pi_{\mathbf{X}_H}^{-1}(\mathbf{x}_H)$ there is a $\mathbf{v}_L \in \pi_{\mathbf{X}_L}^{-1}(\mathbf{x}_L)$ with $\tau(\mathbf{v}_L) = \mathbf{v}_H$ (specifically, any extension of $\mathbf{x}_L$ works).

Now we show the "only if" direction. To show 1: suppose not. Then since $\theta_{V_H}$ is always nonempty there is some $\mathbf{x} \in \mathrm{Val}(\Pi_{X_H})$ such that $\pi_{\mathbf{X}_L \cap \Pi_{X_H}}(\mathbf{x}) = \pi_{\mathbf{X}_L \cap \Pi_{X_H}}(\mathbf{x}_L)$ and $\tau_{X_H}(\mathbf{x}) \neq \pi_{X_H}(\mathbf{x}_H)$. But then we cannot have $\tau(\mathbf{v}_L) \in \pi_{\mathbf{X}_H}^{-1}(\mathbf{x}_H)$ for any $\mathbf{v}_L \in \pi_{\mathbf{X}_L}^{-1}(\mathbf{x}_L) \cap \pi_{\Pi_{X_H}}^{-1}(\mathbf{x})$, which is nonempty. To show 2: suppose not. Then there is some $v_H \in \mathrm{Val}(V_H)$ such that there is no $\mathbf{x} \in \mathrm{Val}(\Pi_{V_H})$ with $\pi_{\mathbf{X}_L \cap \Pi_{V_H}}(\mathbf{x}) = \pi_{\mathbf{X}_L \cap \Pi_{V_H}}(\mathbf{x}_L)$ and $\tau_{V_H}(\mathbf{x}) = v_H$. But then for any $\mathbf{v}_H \in \pi_{\mathbf{X}_H}^{-1}(\mathbf{x}_H) \cap \pi_{V_H}^{-1}(v_H)$, which is nonempty since $V_H \notin \mathbf{X}_H$, there cannot be any $\mathbf{v}_L \in \tau^{-1}(\mathbf{v}_H) \cap \pi_{\mathbf{X}_L}^{-1}(\mathbf{x}_L)$, contradicting that $\tau(\mathbf{x}_L) = \mathbf{x}_H$. $\square$

*Proof of Prop. 2.* Suppose $\mathcal{H}$ is not effective so there is some $\{(\mathbf{y}_H^j)_{\mathbf{x}_H^j}\}_{1 \leq j \leq n}$ in its counterfactual support such that for some $k$ and $Y$ we have that $\pi_Y(\mathbf{y}_H^k) \neq \pi_Y(\mathbf{x}_H^k)$. This means there is some $(\mathbf{y}_L^l)_{\mathbf{x}_L^l}$ in a counterfactual in the support of $\mathcal{L}$ with $\tau(\mathbf{y}_L^l) = \mathbf{y}_H^k$ and $\tau(\mathbf{x}_L^l) = \mathbf{x}_H^k$. By Lem. A.1, 1, there is no $\mathbf{x} \in \mathrm{Val}(\Pi_Y)$ such that $\pi_{\mathbf{Y}_L^l \cap \Pi_Y}(\mathbf{x}) = \pi_{\mathbf{Y}_L^l \cap \Pi_Y}(\mathbf{y}_L^l)$ and $\pi_{\mathbf{X}_L^l \cap \Pi_Y}(\mathbf{x}) = \pi_{\mathbf{X}_L^l \cap \Pi_Y}(\mathbf{x}_L^l)$. This implies that there is some $Y' \in \mathbf{X}_L^l \cap \mathbf{Y}_L^l \cap \Pi_Y$ such that $\pi_{Y'}(\mathbf{x}_L^l) \neq \pi_{Y'}(\mathbf{y}_L^l)$, contradicting the effectiveness of $\mathcal{L}$.

As for the second claim, by induction it suffices to show this for the case where $\mathcal{H}'$ is identical to $\mathcal{H}$ except that it lacks a single potential outcome $Y_\mathbf{x}$. Consider the model $\mathcal{L}'$ formed applying the mapping $\{(\mathbf{y}_L^i)_{\mathbf{x}_L^i}\}_{1 \leq i \leq m} \mapsto \{(\mathbf{z}_L^i)_{\mathbf{x}_L^i}\}_{1 \leq i \leq m}$ of counterfactuals to $\mathcal{L}$, where $\mathbf{z}_L^i = \pi_{\mathbf{Y}_L^i \setminus \Pi_Y}(\mathbf{y}_L^i)$ if $\tau(\mathbf{x}_L^i) = \mathbf{x}$ and $\mathbf{z}_L^i = \mathbf{y}_L^i$ otherwise. It follows from Lem. A.1 that $\mathcal{H}' \prec_\tau \mathcal{L}'$. $\square$

*Proof of Thm. 1.* Let $\mathcal{R}$ be as in Def. 1 and index its potential outcomes $\mathfrak{O}$ as $\{(\mathbf{Y}^i)_{\mathbf{x}^i}\}_{1 \leq i \leq n} \cup \{(\mathbf{Y}^{n+1})_{\mathbf{x}^{n+1}}\}$ where $\mathbf{x}^i \neq \varnothing$ for any $1 \leq j \leq n$ and $\mathbf{x}^{n+1} = \varnothing$; here $\varnothing$ represents the empty intervention on $\mathbf{V}$. We assume that $n > 0$, since otherwise $\mathcal{R}$ is trivially representable. The potential response functions of $\mathcal{R}$ are $\{f_{Y_\mathbf{x}}\}_{Y_\mathbf{x}}$.

Define a low-level set of variables $\mathbf{V}_L = \{(V, j) : {}_{1 \leq j \leq n+1}^{V \in \mathbf{V}}\}$ with $\mathrm{Val}(V, j) = \mathrm{Val}(V) \cup \{\star\}$ where $\star \notin \mathrm{Val}(V)$ for each $(V, j) \in \mathbf{V}_L$. Define $\tau$ as a constructive translation with a partition $\Pi$, defined by $\Pi_V = \{(V, j)\}_{1 \leq j \leq n+1}$ for each $V \in \mathbf{V}$, and $\tau_V(\mathbf{p}_L) = p_j$ iff $\mathbf{p}_L \in \mathrm{Val}(\Pi_V)$ is such that there is exactly one $j$ such that $\pi_{(V,j)}(\mathbf{p}_L) \neq \star$, with $p_j$ this value. Let

$$\mathfrak{O}_L = \left\{(Y, i)_{\mathbf{x}_L^i} : {}_{Y \in \mathbf{Y}^i}^{1 \leq i \leq n}\right\} \cup \left\{(Y, n+1)_{\varnothing_L} : Y \in \mathbf{Y}^{n+1}\right\} \cup \left\{(Y, i)_{\mathbf{x}_L^j} : {}_{Y \in \mathbf{Y}^i}^{\substack{1 \leq i,j \leq n \\ i \neq j}}\right\}$$

$$\cup \left\{(Y, n+1)_{\mathbf{x}_L^i} : {}_{Y \in \mathbf{Y}^{n+1}}^{1 \leq i \leq n}\right\} \cup \left\{(Y, i)_{\varnothing_L} : {}_{Y \in \mathbf{Y}^i}^{1 \leq i \leq n}\right\} \quad \text{(A.1)}$$

where for each $1 \leq i \leq n$, we let $\mathbf{X}_L^i = \{(X, i) : X \in \mathbf{X}^i\}$ with $\pi_{(X,i)}(\mathbf{x}_L^i) = \pi_X(\mathbf{x}^i)$ for every $X \in \mathbf{X}^i$, $\varnothing_L$ is an empty intervention on $\mathbf{V}_L$, and define a set of potential responses $\mathfrak{F}_L = \{f_{Z_\mathbf{x}}^L\}_{Z_\mathbf{x}}$ for each of these outcomes via

$$f_{Z_\mathbf{x}}^L = \begin{cases} f_{Y_{\mathbf{x}^i}}, & Z_\mathbf{x} = (Y, i)_{\mathbf{x}_L^i} \\ f_{Y_{\mathbf{x}^{n+1}}}, & Z_\mathbf{x} = (Y, n+1)_{\varnothing_L} \\ f_\star, & \text{otherwise} \end{cases} \quad \text{(A.2)}$$

where $f_\star$ is a constant function with $f_\star(u) = \star$ for any $u \in \mathcal{U}$.

We claim that $\mathcal{R}_L = \langle \mathcal{U}, \mathbf{V}_L, \mathfrak{O}_L, \{f_{Z_\mathbf{x}}^L\}_{Z_\mathbf{x}}, P\rangle$, where $\mathcal{U}$ and $P$ are the same as those in $\mathcal{R}$, abstracts to $\mathcal{R}$ and is representable. To show the former, we employ the following result:

**Lemma A.2.** Let $\mathbf{x}_L \in \mathrm{Val}(\mathbf{X}_L)$ for some $\mathbf{X}_L \subset \mathbf{V}_L$ and suppose that there is some $\mathbf{X} \subset \mathbf{V}$ and $j \in \{1, \ldots, n+1\}$ such that $\pi_{(V,i)}(\mathbf{x}_L) \neq \star$ iff $V \in \mathbf{X}$ and $i = j$; suppose further that for each $V$, there is at least one $i \in \{1, \ldots, n+1\}$ such that $(V, i) \notin \mathbf{X}_L$ or $\pi_{(V,i)}(\mathbf{x}_L) \neq \star$. Then $\tau(\mathbf{x}_L) = \mathbf{x}_H$ where $\mathbf{X}_H = \mathbf{X}$ and $\pi_X(\mathbf{x}_H) = \pi_{(X,i)}(\mathbf{x}_L)$ for each $X \in \mathbf{X}$.

*Proof.* This follows directly from the construction of $\tau$, in light of Lem. A.1. $\square$

Lem. A.2 implies that $\tau(\varnothing_\mathrm{L}) = \varnothing$, $\tau(\mathbf{x}_\mathrm{L}^i) = \mathbf{x}^i$ for each $i$. Further, for each $u \in \mathcal{U}$ and $i$ it implies that $\tau(\mathbf{y}_i^\mathrm{L}(u)) = \mathbf{y}_i(u)$ where $\mathbf{y}_i^\mathrm{L}(u)$ is the partial setting of $\mathbf{V}_\mathrm{L}$ induced by $\mathcal{R}_\mathrm{L}$ for potential outcomes under $\mathbf{x}_i^i$ for unit $u$, viz., by (A.1) and (A.2), that with the projections $\pi_{(Y,i)}(\mathbf{y}_i^\mathrm{L}(u)) = f_{Y_{\mathbf{x}^i}}(u)$ for each $Y \in \mathbf{Y}^i$, $\pi_{(Y,j)}(\mathbf{y}_i^\mathrm{L}(u)) = \star$ for each $j \neq i$, $Y \in \mathbf{Y}^j$; and $\mathbf{y}_i(u)$ is the analogue in $\mathcal{R}$, with $\pi_Y(\mathbf{y}_i(u)) = f_{Y_{\mathbf{x}^i}}(u)$ for $Y \in \mathbf{Y}^i$. These facts establish that $P_\mathrm{cf}^\mathcal{R} = \tau_*(P_\mathrm{cf}^{\mathcal{R}_\mathrm{L}})$ so that $\mathcal{R} \prec_\tau \mathcal{R}_\mathrm{L}$.

Next we demonstrate that $\mathcal{R}_\mathrm{L}$ is representable by constructing an explicit SCM representation $\mathcal{M}_\mathrm{L} = \langle \{U\}, \mathbf{V}_\mathrm{L}, \{e_{V_\mathrm{L}}\}_{V_\mathrm{L} \in \mathbf{V}_\mathrm{L}}, P \rangle$, where $U$ is an exogenous variable with $\mathrm{Val}(U) = \mathcal{U}$, while $P$ is the same distribution as in $\mathcal{R}$ and $\mathcal{R}_\mathrm{L}$. It will be convenient to set up $\mathcal{M}_\mathrm{L}$ so as to be *recursive*. We say that an $\mathcal{M}$ over endogenous variables $\mathbf{V}$ is recursive if there is a total order $<$ on $\mathbf{V}$ such that $\mathbf{Pa}_V \subset \{V' \in \mathbf{V} : V' < V\}$ for every $V \in \mathbf{V}$. Recursiveness guarantees uniqueness and existence of solutions under every unit and intervention, regardless of the particular structural functions $\{f_V\}_V$ and merely by virtue of their signatures:

**Lemma A.3.** If $\mathcal{M}$ is recursive then there is a unique $\mathbf{v} \in \mathrm{Val}(\mathbf{V})$ such that $\mathcal{M}_\mathbf{x}, \mathbf{u} \vDash \mathbf{v}$ for any $\mathbf{x} \in \bigcup_{\mathbf{X} \subset \mathbf{V}} \mathrm{Val}(\mathbf{X})$, unit $\mathbf{u}$.

*Proof.* Assume without loss $\mathbf{V} = \{V_1, \ldots, V_m\}$ with $V_1 < \cdots < V_m$ and prove by induction, the inductive hypothesis being that there is a unique $\mathbf{v}$ over $\{V_1, \ldots, V_j\}$ for $j \leq m$, which extends as $\pi_{V_{j+1}}(\mathbf{v}) = f_{V_{j+1}}(\mathbf{u}, \pi_{\{V_1, \ldots, V_j\}}(\mathbf{v}))$ if $V_{j+1} \notin \mathbf{X}$ and $\pi_{V_{j+1}}(\mathbf{x})$ otherwise. $\quad\square$

For the recursive order $<$, pick any $<$ such that $(V, h) < (V', h')$ for any $V, V' \in \mathbf{V}$ and $1 \leq h, h' \leq n+1$ whenever $h < h'$, for any $1 \leq j \leq n$, we have that $(X, j) < (Y, j)$ for any $X \in \mathbf{X}^j \setminus \mathbf{Y}^j$, $Y \in \mathbf{Y}^j$, and $(Y_l, n+1) < (Y_{l'}, n+1)$ whenever $l < l'$, where we fix an arbitrary indexing $\{Y_l : 1 \leq l \leq |\mathbf{Y}^{n+1}|\}$ of $\mathbf{Y}^{n+1}$. Let $\mathbf{Pa}_{V_\mathrm{L}} = \{V' \in V_\mathrm{L} : V' < V_\mathrm{L}\}$, $\mathbf{U}_{V_\mathrm{L}} = \mathbf{U}$ for each $V_\mathrm{L}$ and define the structural function $e_{(V,h)}$ by

$$
e_{(V,h)}(u, \mathbf{p}) = \begin{cases} f_{V_\varnothing}(u), & h = n+1, V \in \mathbf{Y}^{n+1}, \mathbf{p} \text{ is such that } \pi_{(Y_{l'}, n+1)}(\mathbf{p}) = f_{V_\varnothing}(u) \\ & \quad \text{for any } l' \text{ such that } n+1 \leq l' < l \text{ and} \\ & \quad \pi_{(V', h')}(\mathbf{p}) = \star \text{ for any other } (V', h') \\ f_{V_{\mathbf{x}^h}}(u), & 1 \leq h \leq n, V \in \mathbf{Y}^h, \left|\mathbf{X}^h \setminus \mathbf{Y}^h\right| > 0, \mathbf{p} \text{ is such that} \\ & \quad \pi_{(X,h)}(\mathbf{p}) = \pi_X(\mathbf{x}_h) \text{ for any} \\ & \quad X \in \mathbf{X}^h \setminus \mathbf{Y}^h \text{ and } \pi_{(V', h')}(\mathbf{p}) = \star \text{ for any other } (V', h') \\ \star, & \text{otherwise} \end{cases}
$$
(A.3)

We can find the unique solution of $\mathcal{M}_\mathrm{L}$ under any of our interventions of interest by following these equations in order:

**Lemma A.4** (Solutions of $\mathcal{M}_\mathrm{L}$). Let $\mathbf{x} \in \{\mathbf{x}_\mathrm{L}^j\}_{1 \leq j \leq n} \cup \{\varnothing_\mathrm{L}\}$ and $i \in \mathcal{U}$. Then $(\mathcal{M}_\mathrm{L})_\mathbf{x}, i \vDash \mathbf{v} \in \mathrm{Val}(\mathbf{V}_\mathrm{L})$ where

$$
\pi_{(V,h)}(\mathbf{v}) = \begin{cases} \pi_V(\mathbf{x}_j) & \mathbf{x} = \mathbf{x}_\mathrm{L}^j \text{ for some } j, V \in \mathbf{X}^j \setminus \mathbf{Y}^j, h = j \\ f_{V_{\mathbf{x}^j}}(i) & \mathbf{x} = \mathbf{x}_\mathrm{L}^j \text{ for some } j, V \in \mathbf{Y}^j, h = j \\ f_{V_\varnothing}(i) & \mathbf{x} = \varnothing_\mathrm{L}, h = n+1, V \in \mathbf{Y}^{n+1} \\ \star & \text{otherwise} \end{cases}
$$
(A.4)

*Proof.* Suppose $\mathbf{x} = \mathbf{x}_\mathrm{L}^j$ for some $j \leq n$. We claim that $\pi_{(V,k)}(\mathbf{v}) = \star$ for any $V$ and $k < j$. The only way it could be otherwise is in the second case of (A.3), but since $k \neq j$ and $\left|\mathbf{X}^k \setminus \mathbf{Y}^k\right| > 0$ there is here at least one $(X, k) < (Y^k, k)$ with $\pi_{(X,k)}(\mathbf{v}) = \star$, so that this case in fact cannot apply. Now for $k = j$, it is clear by construction of $\mathbf{x}_\mathrm{L}^j$ that $\pi_{(V,j)}(\mathbf{v}) = \pi_V(\mathbf{x}^j)$ for each $V \in \mathbf{X}^j \setminus \mathbf{Y}^j$. For $V \in \mathbf{Y}^j$, if $V \in \mathbf{X}^j$ then by effectiveness and the construction of $\mathbf{x}_\mathrm{L}^j$, we have that $\pi_{(V,j)}(\mathbf{v}) = \pi_V(\mathbf{x}^j) = f_{V_{\mathbf{x}^j}}(i)$. Otherwise, since $\mathbf{X}^j \neq \varnothing$ and given what we have already found about $\mathbf{v}$ we fall into the second case in (A.3). Thus $\pi_{(V,j)}(\mathbf{v}) = f_{V_{\mathbf{x}^j}}(i)$. For $k > j$, if $k \leq n$ then the second case of (A.3) cannot apply so $\mathbf{v}$ extends only by $\star$'s, and if $k = n+1$ then since $\mathbf{x}^j \neq \varnothing$ the first case of (A.3) likewise cannot apply, and we get only $\star$'s. This gives us precisely the $\mathbf{v}$ specified by the first, second, and fourth cases in (A.4).

Next suppose $\mathbf{x} = \varnothing_{\mathrm{L}}$. Then $\pi_{(V,h)}(\mathbf{v}) = \star$ for any $V$ and $h < n+1$: it could only be otherwise in the second case of (A.3) where since $\left|\mathbf{X}^h \setminus \mathbf{Y}^h\right| > 0$ there is here at least one $(X,h) < (V,h)$ with $\pi_{(X,h)}(\mathbf{v}) = \star$ by the third case of (A.3). Now if $h = n+1$ then we have $\pi_{(Y_1,n+1)}(\mathbf{v}) = f_{V_\varnothing}(i)$ by the first case of (A.3) and (inductively) likewise for $Y_2, \ldots, Y_{|\mathbf{Y}^{n+1}|}$. For any $V \notin \mathbf{Y}^{n+1}$, we have $\pi_{(V,n+1)}(\mathbf{v}) = \star$. We thus obtain exactly the last two cases of (A.4). $\qquad\square$

To show that $p_{\mathrm{cf}}^{\mathcal{M}_{\mathrm{L}}}$ marginalizes to $p_{\mathrm{cf}}^{\mathcal{R}_{\mathrm{L}}}$, because $\mathcal{M}_{\mathrm{L}}$ and $\mathcal{R}_{\mathrm{L}}$ are constructed so as to share the same set of units and distribution over them, it suffices to show that for each $i \in \mathcal{U}$ we have that $(\mathcal{M}_{\mathrm{L}})_{\mathbf{x}}, i \models (V,h) = f_{(V,h)_{\mathbf{x}}}^{\mathrm{L}}(i)$ for every $(V,h)_{\mathbf{x}} \in \mathfrak{O}_{\mathrm{L}}$. This is easily seen by inspecting and comparing (A.4), (A.1), and (A.2). $\qquad\square$

**SUTVA**

The following remarks are in the context of the example from the discussion after Def. 9 (see also that after Thm. 1).

**Remark A.1.** We show how our framework can model unit-wise treatment assignment and the first part of SUTVA as making a particular abstraction viable.

Consider a model $\langle \{s\}, \{U, Y, T\}, \mathfrak{O}, \mathfrak{F}, \cdot \rangle$ where $s$ is some singleton element and $U$ is an endogenized unit variable (so that the exogenous unit set $\mathcal{U}$ is trivial in this model), $Y$ encodes survival with $\mathrm{Val}(Y) = \{0,1\}$, and $T$ is a treatment variable that encodes whether *each* unit was assigned to treatment, with $\mathrm{Val}(T) = \bigtimes_{u \in \mathrm{Val}(U)} \mathrm{Val}(Y)$. The list $\mathfrak{O}$ contains the $2^{|\mathrm{Val}(U)|} |\mathrm{Val}(U)|$ potential responses $Y_{\mathbf{x}}$ for each possible $\mathbf{x}$ where $\pi_T(\mathbf{x}) = \mathbf{u} \in \bigtimes_{u \in \mathrm{Val}(U)} \{0,1\}$ and $\pi_U(\mathbf{x}) = u \in \mathrm{Val}(U)$, and $\mathfrak{F}$ maps this to the survival outcome for patient $u$ when the vector $\mathbf{u}$ encodes treatment assignments for each patient. Under the above SUTVA assumption, this outcome depends only on the component $\pi_u(\mathbf{u})$ of $\pi_T(\mathbf{x})$ corresponding to $U = u$. This means that a constructive abstraction $\tau$ to high-level variables $\{T', Y\}$ where $\mathrm{Val}(T') = \mathrm{Val}(U) \times \{0,1\}$ with partition $\Pi$ is viable: $\Pi_{T'} = \{T, U\}$, $\Pi_Y = \{Y\}$, and $\tau(\mathbf{x}) = \big(\pi_U(\mathbf{x}), \pi_{\pi_U(\mathbf{x})}(\pi_T(\mathbf{x}))\big)$. In turn, given a distribution on units, we can exogenize them out of $T'$, showing that this model is equivalent to the familiar one with binary treatment and potential outcomes $Y_t(u)$.

**Remark A.2.** We show how to model several alternative ways mentioned by [5] of making the second part of SUTVA hold.

The first way is for each unit to receive only one treatment level; then the possibility of inconsistency in abstraction is excluded. Our framework requires that the potential responses are total functions (with domain the set of units) and therefore that the outcome for each unit and treatment level is defined; nevertheless, it is possible to redo the analysis from the beginning up to causal abstraction, modifying Def. 1 to admit partial potential response functions. One would then find that this version of SUTVA guarantees the viability of the abstraction collapsing all treatment levels.

The second way is to admit stochastic treatments. Suppose we have two treatment levels $\mathrm{tr}^i$, $\mathrm{tr}^j$ and some $u \in \mathcal{U}$, occurring with probability $P(u)$, for which $Y_{\mathrm{tr}^i}(u) \neq Y_{\mathrm{tr}^j}(u)$. Then for any $p^i, p^j \geq 0$ with $p^i + p^j = 1$ we can form a new model with $\mathrm{tr}^i$, $\mathrm{tr}^j$ merged into a single treatment $\mathrm{tr}$, but $u$ split into two new units $u^i, u^j$ with respective probabilities $p^i P(u), p^j P(u)$. In this model we define $Y_{\mathrm{tr}}(u^i) = Y_{\mathrm{tr}^i}(u)$, $Y_{\mathrm{tr}}(u^j) = Y_{\mathrm{tr}^j}(u)$, and the violation to the condition in question is removed since we end up with only a single treatment level $\mathrm{tr}$. Further, the expectation of $Y_{\mathrm{tr}}$ conditioned on drawing $u^i$ or $u^j$ is equal to the weighted average of $Y_{\mathrm{tr}^i}(u)$ and $Y_{\mathrm{tr}^j}(u)$. This is really a different definition in which we mix incompatible outcomes rather than consider them to invalidate the entire abstraction—study of such a notion may prove fruitful, but is outside the scope of this article.

A final way to make this condition hold is to coarsen the outcome, by, e.g. blurring multiple health statuses into the binary distinction between surviving and not. This can in fact already be seen in our framework. Let $\tau$ be the abstraction from the running example coarsening treatment levels (values of $T$), $\tau'$ be that coarsening the outcome (values of $Y$), and $\mathcal{L}$ be our lowest-level model. Then it is possible that we have some $\mathcal{H}', \mathcal{H}$ for which $\mathcal{H} \prec_\tau \mathcal{H}' \prec_{\tau'} \mathcal{L}$, but no $\mathcal{H}$ for which $\mathcal{H} \prec_\tau \mathcal{L}$. In this sense (by moving to the already coarser model $\mathcal{H}'$) the second part of SUTVA can be made to hold.

# B  Inference (§2)

**Proposition B.1.** $(6), (7), (8) \models \text{LATE} = \text{ITT}_1/\text{ITT}_2$.

*Proof.* Regarding the encoding of expected values: for any function $f : \text{Val}_{\text{cf}} \to \mathbb{R}$ we can define an expected value $\mathbf{E_u}(f) = \sum_{\mathbf{u} \in \mathcal{U}} P(\mathbf{u}) f(\mathfrak{O}(\mathbf{u}))$, abbreviated $\mathbf{E}(f)$. If $f$ is integer-valued then $\mathbf{E}(f)$ is expressible within our formal language. Thus LATE is equal to a ratio[2] of our terms,

$$\text{LATE} = \mathbf{E}(Y_{x^+} - Y_{x^-} | x_{z^+}^+ \wedge x_{z^-}^-) = \frac{\mathbf{P}(y_{x^+}^+ \wedge y_{x^-}^- \wedge x_{z^+}^+ \wedge x_{z^-}^-) - \mathbf{P}(y_{x^+}^- \wedge y_{x^-}^- \wedge x_{z^+}^+ \wedge x_{z^-}^-)}{\mathbf{P}(x_{z^+}^+ \wedge x_{z^-}^-)}.$$

We show $\text{LATE} = \text{ITT}_1/\text{ITT}_2$; (6) shows the second term of $\text{ITT}_2$ vanishes, while the first term is the denominator of LATE, so it suffices to show equality of the numerator and $\text{ITT}_1$. All of the following arithmetic moves rely only on AX. Note that

$$\mathbf{P}(y_{z^+, X_{z^+}}^+ \wedge y_{z^-, X_{z^-}}^-) = \sum_{x, x'} \mathbf{P}(y_{z^+, x}^+ \wedge x_{z^+} \wedge y_{z^-, x'}^- \wedge x_{z^-}') = \sum_{x, x'} \mathbf{P}(y_x^+ \wedge x_{z^+} \wedge y_{x'}^- \wedge x_{z^-}')$$

the last equality following from (8) and (7). Any terms for $x = x'$ vanish by (7) (using also that $\mathbf{P}(\epsilon) = 0 \to \mathbf{P}(\epsilon \wedge \zeta) = 0$ is derivable from basic probabilistic logic), the term for $x = x^-$ and $x' = x^+$ vanishes by (6), and we are left with exactly the first term of the numerator. We can analogously derive equality of the expressions with negative coefficients. $\square$

**Proposition B.2.** Under representability, $\text{ITT}_1 = \mathbf{E}(Y_{z^+} - Y_{z^-})$.

*Proof.* We claim that $\text{ITT}_1 = \mathbf{E}(Y_{z^+, X_{z^+}} - Y_{z^-, X_{z^-}}) = \mathbf{E}(Y_{z^+} - Y_{z^-})$ and thus that

$$\mathbf{P}(y_{z^+, X_{z^+}}^+ \wedge y_{z^-, X_{z^-}}^-) - \mathbf{P}(y_{z^+, X_{z^+}}^- \wedge y_{z^-, X_{z^-}}^+) = \mathbf{P}(y_{z^+}^+ \wedge y_{z^-}^-) - \mathbf{P}(y_{z^+}^- \wedge y_{z^-}^+)$$

under representability. It suffices to show $\text{Rep.} \models y_{z, X_z} \leftrightarrow y_z$ where Rep. stands for representability. Note that $\text{Rep.} \models y_{z, X_z} \leftrightarrow \bigvee_{x, y'} y_{zx} \wedge x_z \wedge y_z' \leftrightarrow \bigvee_x y_{xz} \wedge x_z \wedge y_z$, the last step since $(2) \models x_z \wedge y_z' \to y_{xz}'$ contradicting $y_{xz}$ when $y' \neq y$. Similarly by setting $y' = y$, we have $\text{Rep.} \models \bigvee_x y_{xz} \wedge x_z \wedge y_z \leftrightarrow \bigvee_x x_z \wedge y_z \leftrightarrow y_z$. Note that no additional outcomes are used here except $Y_z$, which is necessary to state the result. $\square$

*Proof of Thm. 2.* Soundness is straightforward. For completeness, show that any consistent formula $\varphi$ is satisfiable. By the completeness proof for AX [3, Thm. 6] we get a distribution $P$ satisfying $\varphi$ over joint valuations $\text{Val}(\mathfrak{O}_\varphi)$, where $\mathfrak{O}_\varphi$ is the set of potential outcomes appearing in $\varphi$. To construct a satisfying model $\mathcal{R}$, add a unit $u$ that witnesses any $\mathbf{o} \in \text{Val}(\mathfrak{O}_\varphi)$ for which $P(\mathbf{o}) > 0$. It is clear that $\mathcal{R} \in \mathfrak{R}(S)$ since $\varphi$ is consistent with $\text{T}(S)$. $\square$

*Proof of Cor. 1.* Observe that $\mathfrak{R}_{\text{eff}} = \mathfrak{R}(1)$. $\square$

*Proof of Thm. 3.* Soundness has been shown in prior works [7], and in any case is straightforward ($\text{ER}^{\mathcal{G}}$ since each variable can only be a function of its parents in $\mathfrak{M}(\mathcal{G})$ and $\text{cf-sep}^{\mathcal{G}}$ since the functional mechanisms determining unconfounded nodes must be independent; see also Cor. B.1 below for a derivation from d-separation).

Completeness: our proof is a modification of the proof for $\text{AX}_3$ in [3, Thm. 6]. We take the same steps as in that proof, the strategy being to show that any consistent $\varphi$ is satisfiable, up to and including the normal form in Lem. 8, except that we use the full set $\mathbf{V}$ in place of $\mathbf{V}_\varphi$ (this is possible since $\mathbf{V}$ is finite in our case).

Let $\mathcal{C}$ be the set of connected components under the edge relation $\leftarrow\!\text{--}\!\rightarrow$ of $\mathcal{G}$. Each $\mathbf{C} \in \mathcal{C}$ is a subset $\mathbf{C} \subset \mathbf{V}$ that is a complete graph under $\leftarrow\!\text{--}\!\rightarrow$, by assumption. For each $\mathbf{C}$ let

$$\Delta_{\mathbf{C}} = \left\{ \bigwedge_{\substack{C \in \mathbf{C} \\ \mathbf{p}_C \in \text{Val}(\mathbf{Pa}_C^{\mathcal{G}})}} C_{\mathbf{p}_C} = c_{C, \mathbf{p}_C} : \substack{c_{C, \mathbf{p}_C} \in \text{Val}(C) \text{ for each } C \in \mathbf{C}, \\ \mathbf{p}_C \in \text{Val}(\mathbf{Pa}_C^{\mathcal{G}})} \right\}$$

---

[2] The ratio $\mathbf{t}_1 = {^{\mathbf{t}_2}}/{_{\mathbf{t}_3}}$ is an abbreviation for $\mathbf{t}_1 \cdot \mathbf{t}_3 = \mathbf{t}_2 \in \mathcal{L}$. Likewise in Ex. 4.

be the collection of counterfactuals describing complete functional mechanisms for each $C \in \mathbf{C}$. The notation $C_{\mathbf{p}_C} = c_{C,\mathbf{p}_C}$ represents the outcome that $C = c_{C,\mathbf{p}_C}$ under the intervention $\mathbf{p}_C$; we use such notation in order to be explicit about the choice of variable $C$. The below is our analogue of [3, Lem. 8].

**Lemma B.1.** Let $\varphi$ be a conjunction of literals. Then there are polynomial terms $\{\mathbf{t}_i, \mathbf{t}'_{i'}\}_{i,i'}$ in the variables $\big\{ \mathbf{P}(\delta_{\mathbf{C}}) : {}^{\delta_{\mathbf{C}} \in \Delta_{\mathbf{C}}}_{\mathbf{C} \in \mathcal{C}} \big\}$ such that

$$\mathrm{T}(\mathcal{G}) \vdash \varphi \leftrightarrow \bigwedge_{\substack{\delta_{\mathbf{C}} \in \Delta_{\mathbf{C}} \\ \mathbf{C} \in \mathcal{C}}} \mathbf{P}(\delta_{\mathbf{C}}) \geq 0 \wedge \bigwedge_{\mathbf{C} \in \mathcal{C}} \Big[ \sum_{\delta_{\mathbf{C}} \in \Delta_{\mathbf{C}}} \mathbf{P}\left(\delta_{\mathbf{C}}\right) = 1 \Big] \wedge \bigwedge_i \mathbf{t}_i \geq 0 \wedge \bigwedge_{i'} \mathbf{t}'_{i'} > 0. \quad \text{(B.1)}$$

*Proof.* This will follow from the fact that we have the below (B.2) for any $\delta \in \Delta_{\mathrm{sat}}$. Here $\Delta_{\mathrm{sat}}$ is the set defined in [3] as the satisfiable subset of complete interventional state descriptions $\Delta = \big\{ \bigwedge_{\substack{\mathbf{x} \in \mathrm{Val}(\mathbf{X}) \\ \mathbf{X} \subset \mathbf{V}}} \bigwedge_{V \in \mathbf{V}} V_{\mathbf{x}} = \pi_V(\mathbf{v_x}) : {}^{\mathbf{v_x} \in \mathrm{Val}(\mathbf{V})}_{\text{for each } \mathbf{x}} \big\}$.

$$\mathrm{T}(\mathcal{G}) \vdash \mathbf{P}(\delta) = \prod_{\mathbf{C} \in \mathcal{C}} \mathbf{P}\left( \bigwedge_{\substack{C \in \mathbf{C} \\ \mathbf{p}_C \in \mathrm{Val}(\mathbf{Pa}_C^{\mathcal{G}})}} C_{\mathbf{p}_C} = c_{C,\mathbf{p}_C} \right) \quad \text{(B.2)}$$

where $c_{C,\mathbf{p}_C} \in \mathrm{Val}(C)$ for each $C$ is the outcome of $C$ in $\delta$ in the outer conjunct corresponding to $\mathbf{x} = \mathbf{p}_C$ and the inner conjunct corresponding to $V = C$. To see (B.2), note that for each $V \in \mathbf{V}$ we can remove all conjuncts in a $\delta$ for any $\mathbf{x}$ that do not correspond to a setting $\mathbf{p}_C$ of the parents $\mathbf{Pa}_C^{\mathcal{G}}$ by $\mathsf{ER}^{\mathcal{G}}$ and composition, which we can use since $\mathsf{SCM}$ is part of $\mathrm{T}(\mathcal{G})$. We thereby obtain $\mathbf{P}(\delta) = \mathbf{P}\big( \bigwedge_{\substack{V \in \mathbf{V} \\ \mathbf{p} \in \mathbf{Pa}_V^{\mathcal{G}}}} V_{\mathbf{p}} = \pi_V(\mathbf{v_p}) \big)$ and this gives us the $c_{V,\mathbf{p}} = \pi_V(\mathbf{v_p})$. Next, we apply $\mathsf{cf\text{-}sep}^{\mathcal{G}}$ to this expression, giving us the final product in (B.2) factorizing over all $\mathbf{C} \in \mathcal{C}$. $\square$

Now, as in the proof of [3, Thm. 6], the Positivstellensatz shows that consistency of $\varphi$ implies there is a real solution to (B.1) for $\{\mathbf{P}(\delta_{\mathbf{C}})\}_{\substack{\delta_{\mathbf{C}} \in \Delta_{\mathbf{C}} \\ \mathbf{C} \in \mathcal{C}}}$. We construct a model $\mathcal{M}$ with diagram $\mathcal{G}$ inducing these probabilities and thereby satisfying $\varphi$. Note that any $\delta_{\mathbf{C}} \in \Delta_{\mathbf{C}}$ defines a function $f_C^{\delta_{\mathbf{C}}} : \mathrm{Val}(\mathbf{Pa}_C^{\mathcal{G}}) \to \mathrm{Val}(C)$ for each $C \in \mathbf{C}$ via setting $f_C^{\delta_{\mathbf{C}}}(\mathbf{p}_C)$ to the value $C$ is set to under $\mathbf{p}_C$ in $\delta_{\mathbf{C}}$. We let $\mathcal{M} = \langle \{U_{\mathbf{C}}\}_{\mathbf{C} \in \mathcal{C}}, \mathbf{V}, \{f_V\}_V, P \rangle$ where $P$ factors as $P(\{u_{\mathbf{C}}\}_{\mathbf{C} \in \mathcal{C}}) = \prod_{\mathbf{C} \in \mathcal{C}} P(u_{\mathbf{C}})$ and for any variable $C \in \mathbf{V}$ with $C \in \mathbf{C}$, we define parent sets $\mathbf{Pa}_C = \mathbf{Pa}_C^{\mathcal{G}}$ and $\mathbf{U}_C = \{U_{\mathbf{C}}\}$; this implies that $\mathcal{M}$ has diagram $\mathcal{G}$ (there is a bidirected edge between any $C, C'$ in the same $\mathbf{C}$, and the statement assumes $\mathcal{G}$'s bidirected arcs form a disjoint union of complete graphs). Finally let $\mathrm{Val}(U_{\mathbf{C}}) = \Delta_{\mathbf{C}}$ for each $\mathbf{C}$, and let $P(\delta_{\mathbf{C}})$ be the probability $\mathbf{P}(\delta_{\mathbf{C}})$ from our solution to (B.1); define the structural functions by $f_C(\mathbf{p}_C, \delta_{\mathbf{C}}) = f_C^{\delta_{\mathbf{C}}}(\mathbf{p}_C)$ for each $C \in \mathbf{C} \in \mathcal{C}$. By construction, it is clear that $\mathcal{M}$ induces exactly the probabilities from our solution to (B.1). $\square$

The following definition of d-separation and its extension to mixed diagrams are standard (e.g., [6, 8]). In a directed graph $\mathcal{D}$ over nodes $\mathbf{V}$, let $\mathbf{De}_V^{\mathcal{D}} = \{V' \in \mathbf{V} : V \to \cdots \to V'\}$ be the set of *descendants* of $V \in \mathbf{V}$, the transitive closure of parenthood, with $V \in \mathbf{De}_V^{\mathcal{D}}$.

**Definition B.1** (Directed d-separation). Given directed graph $\mathcal{D}$, the nodes $X$ and $Y$ are *d-separated* given a set of nodes $\mathbf{Z}$ if every path (consisting of arrows of either direction) from $X$ to $Y$ either has a *collider* node (to be defined shortly) on the path $M$ such that $\mathbf{De}_M^{\mathcal{D}} \cap \mathbf{Z} = \varnothing$, or a non-collider node $M'$ on the path such that $M' \in \mathbf{Z}$. A collider is a vertex $M$ on the path such that the path is of the form $X \rightleftharpoons \cdots \to M \leftarrow \cdots \rightleftharpoons Y$ (the symbol $\rightleftharpoons$ denoting an arrow of either direction). The subsets $\mathbf{X}, \mathbf{Y}$ of nodes are d-separated given $\mathbf{Z}$ if every $X \in \mathbf{X}$ and $Y \in \mathbf{Y}$ are d-separated given $\mathbf{Z}$.

**Definition B.2** (Mixed d-separation). Given mixed causal diagram $\mathcal{G}$ over $\mathbf{V}$, let $\mathcal{C}$ be the set of maximal cliques in $\mathcal{G}$ under its edge relation $\leftarrow\!-\!-\!-\!\to$. Thus every $\mathbf{C} \in \mathcal{C}$ is a subset $\mathbf{C} \subset \mathbf{V}$ of vertices such that $C \leftarrow\!-\!-\!-\!\to C'$ for every $C, C' \in \mathbf{C}$, and there is no $\mathbf{C}' \supsetneq \mathbf{C}$ with this property. Then form the directed graph $\mathcal{D}^{\mathcal{G}}$ with nodes $\mathbf{V} \cup \mathcal{C}$ and edges

$$\{V \to V' : V, V' \in \mathbf{V}, V \to V' \text{ in } \mathcal{G}\} \cup \{\mathbf{C} \to V : \mathbf{C} \in \mathcal{C}, V \in \mathbf{V}, V \in \mathbf{C}\}.$$

Finally, given $\mathbf{X}, \mathbf{Y}, \mathbf{Z} \subset \mathbf{V}$, we say that $\mathbf{X}$ and $\mathbf{Y}$ are d-separated in $\mathcal{G}$ given $\mathbf{Z}$ if the same holds in $\mathcal{D}^{\mathcal{G}}$ (according to Def. B.1).

**Corollary B.1.** Let $\mathsf{d\text{-}sep}^{\mathcal{G}}$ be an encoding of all the conditional independences implied by the graphical d-separation criterion over the diagram $\mathcal{G}$ (including over counterfactuals, where these are determined by parallel networks). Then the system $\mathrm{T}(\mathsf{ER}^{\mathcal{G}}) + \mathsf{d\text{-}sep}^{\mathcal{G}} + \mathsf{SCM}$ is sound and complete under the same assumptions as Thm. 3.

*Proof.* For soundness, see [9] or any standard discussion of d-separation, e.g., that in [7] or [8]. For completeness, in light of Thm. 3 it suffices to show that any instance of $\mathsf{cf\text{-}sep}^{\mathcal{G}}$, as in Def. 13, is an instance of $\mathsf{d\text{-}sep}^{\mathcal{G}}$. Consider the parallel network $\mathcal{G}'$ with $n + n'$ copies of $\mathcal{G}$ and the two sets of nodes $\mathbf{Y} = \{(Y_i)_{\mathbf{P}_i}\}_{1 \leq i \leq n}$, $\mathbf{Y}' = \{(Y_i')_{\mathbf{P}_i'}\}_{1 \leq i \leq n'}$. Since there is no $Y \in \mathbf{Y}$, $Y' \in \mathbf{Y}'$ for which $Y = Y'$ or $Y \leftarrow\text{-}\text{-}\text{-}\text{-}\rightarrow Y'$ in $\mathcal{G}$ and all ingoing edges to $Y$ and $Y'$ have been severed in $\mathcal{G}'$, the only possibility for a (direction-agnostic) path between some such $Y, Y'$ is a bidirected path of length $\geq 2$ or a bidirected path between proper descendants of $Y, Y'$. All such paths contain colliders, so that d-separation delivers the (unconditional) independence (9). $\qquad\square$