# OpenReview forum: "Comparing Causal Frameworks: Potential Outcomes, Structural Models, Graphs, and Abstractions"
_NeurIPS.cc/2023/Conference — NeurIPS 2023 poster_

### Official Review · Reviewer_Qo71 · 2023-06-29

**Soundness:** 3 good
**Presentation:** 3 good
**Contribution:** 3 good
**Rating:** 6
**Confidence:** 3

**Summary:**

In this paper, the authors compare the causal frameworks of the Rubin causal model (RCM) and the structural causal model (SCM) from a logical perspective. This is a pretty theoretical work in which RCMs are connected to SCMs using the notion of *representability* which describes whether an RCM can be represented by an SCM. Utilizing causal abstractions, it is shown that there exists a representable low-level RCM for any RCM. Using a neutral formal language, the authors also model assumption for causal inference which can be applied to both RCMs and SCMs. The authors highlight how related both frameworks are and do not state a general preference towards one or the other.

**Strengths:**

1. This paper is a good contribution towards a theoretical comparison of RCMs and SCMs, comparing them on a very precise, fundamental level.

2. The examples included in the paper are nice and are helpful for understanding (but also see point 2 in weakness section).

3. The writing is very clear and concise.

4. I like that a neutral language is used, giving no preference to either framework.

**Weaknesses:**

1. While I acknowledge the page limit for the submission, the paper would have benefited from some additional examples and clarifications for an easier and faster understanding. Maybe the appendix would be a good place for that.

2. There are several parts in the text where the notation could have been explained better and is not easily understandable (see "Minor Criticism and Comments").

3. Results about the relationship between RCMs and SCMs included in this paper are not very surprising to me, however, the theoretical framework is still very useful, so this should not be seen as a major weakness.

### Minor Criticism and Comments

- Line 53: It should say here that "SUTVA" means "Stable Unit Treatment Value Assumption"
- Line 77: What is $B$?
- Line 121, also later: "Ex. 2" does not link to "Example 2"
- Line 197: What is the inverse of a projection, i. e. $\pi^{-1}$? Intuitively, it makes sense to me but I think it could be explained more clearly
- Line 200, 201: $m$ and $n$ are used without explanation
- Line 203: I understand the idea behind $\tau^{-1}$ but I think a clear definition would be useful
- Line 283: What is $Z$? As in what is the meaning of $Z$? Is it the treatment assignment, with $X$ being whether treatment is applied?
- **Please check** I think the monotonicity assumption (Equation 6) is not defined correctly. It depends on what on how exactly to interpret $X$ and $Z$ but this definition looks not correct to me. I read it as "If an individual gets assigned treatment and takes it, it follows that this individual would also take treatment if it was not assigned"
- Line 304: Please state what "ITT" stands for ("Intention-To-Treat" I assume)
- Lines 347, 348: Should it not be $\mathbf{Pa}_Y^\mathcal{G}$ instead of $\mathbf{Pa}_V^\mathcal{G}$?

**Questions:**

 1. As far as I understand it, the SCM definition only covers hard interventions (Definition 4). How would the extension to soft interventions for SCMs change further results of the paper?

2. I do not understand the sentence in line 354/355. How can equation 6 not be valid if $Z \rightarrow X$. Do we not assume that $Z \rightarrow X$ (that the treatment assignment has an effect on whether treatment is applied)? Or is there another mistake in my understanding? Overall, this sentence is not clear to me.

3. While not the scope of this paper, I would be interested in the implications of the differences, which were hinted at. For example, what kind of "insights that arise when using each that are less transparent when using the other"? Could a language as in this paper help make insights more transparent in the framework in which they would be seen as not as transparent? And more generally speaking, how do these results help us use causal frameworks "better"?

**Limitations:**

No concerns on this.

---

> ### Author Rebuttal · Authors · 2023-08-10
>
> We thank the reviewer for their meticulous comments.
>
> We appreciate the call for more examples and clarifications and have added several new examples, with discussion, to the appendix. Please see also our "global" response to all reviewers.
>
> Concerning the minor criticism and comments:
> - Line 53: thanks for the catch. We have moved the full explanation of this abbreviation from line 247 to 53.
> - Line 77: $B$ is an arbitrary indexing set for the family mentioned on line 77. We have now made this explicit.
> - Line 121: we have fixed the link.
> - Line 197: the ${}^{-1}$ refers to the inverse image, a notion defined for mappings in general (the same applies regarding $\tau^{-1}$ in your comment about line 203). To make this maximally transparent, we have now defined inverse images in Section 1.1 ("Preliminaries").
> - Line 200, 201: $m$, $n$ abbreviate the cardinalities $|O_{\mathrm{L}}|$ and $|O_{\mathrm{H}}|$ of low- and high-level outcome sets respectively. We have added an explanation.
> - Line 203: see response to comment about Line 197
> - Line 283: yes. To make this clearer, we have added an explanation of the meaning of $X$, $Z$, which are indeed interpreted as treatment and prescribed treatment respectively, and elaborated on our explanation of monotonicity ("no defiers," line 281).
> - "Please check": Indeed, $z^+$ and $z^-$ should be swapped in equation (6), and we have done this for the final version.
> - Line 304: we have added an explanation.
> - Line 347, 348: we have fixed these typos.
>
> Responses to questions:
> 1. Perhaps surprisingly, it is possible to show that soft interventions are fully reducible to hard interventions within our language $\mathcal{L}$ (in the sense that soft effects are equivalent to distributions on the results of hard effects). Thus, including such soft interventions would not change our results to the extent that they are couched in $\mathcal{L}$. If the reviewer feels it would be important to mention this, we would be happy to add a footnote (and short accompanying appendix subsection) clarifying this.
> 2. We mean that there is no $\mathcal{G}$ including the edge $Z \rightarrow X$ such that $\mathrm{T}(6)$ is valid over $\mathfrak{M}(\mathcal{G})$, i.e., is true in all SCMs with graph $\mathcal{G}$. Since, as you say, we do indeed assume that $Z \rightarrow X$, this means that monotonicity (6) is not implied by any acceptable graph. Thus it is not a graphical assumption—its source must be fundamentally extra-graphical, contrasting it with, e.g., the exclusion restrictions. We have inserted an additional sentence making this clear.
> 3. We really appreciate this question and think it's an important point for connecting this theoretical work with more practical issues in causal inference. We believe the framework in the paper offers a distinctive—and notably, objective—perspective on the comparison. By framing causal inference in terms of derivability in $\mathcal{L}$, this can help illuminate inferential relationships among assumptions and possible conclusions. Examples of this are the observations in the paper about what exactly is necessary for the well-known LATE derivation, as well as our graphical completeness result. But we definitely don't want to suggest that the framework is unique in its promise for such clarification. The important body of research on SWIGs and related topics (cf. the discussion with Reviewer vtiC) has many other examples of shifts in transparency afforded by news ways of packaging and formulating existing ideas and results, which can in turn spur new results. We are optimistic that the (meta-)framework in the present work will continue to be an important contributor to this wider endeavor.

---

> > ### Comment · Reviewer_Qo71 · 2023-08-12
> > **Thank you for your response**
> >
> > I would like to thank the authors for their response. I will maintain my rating.

---

### Official Review · Reviewer_Ef3C · 2023-07-06

**Soundness:** 4 excellent
**Presentation:** 4 excellent
**Contribution:** 4 excellent
**Rating:** 8
**Confidence:** 4

**Summary:**

The paper offers a mathematically rigorous comparison of the potential outcomes and structural causal models frameworks for causality. Their comparison goes further than previous work, in part by invoking the idea of an abstraction, and in part by invoking recent  axiomiatizations for probabilistic SCMs. As a result, they offer several new and important connections between the two frameworks, as well as offering novel insights into the frameworks separately.

**Strengths:**

Given the importance of both the RCM and SCM frameworks, and the long-standing debate as to how exactly they relate, these novel results about their relation are extremely valuable. This paper does so much in such little space, that I am confident its results will have an impact on many different issues within the causality literature. I believe this paper is an important milestone when it comes to the mathematical foundations of causal frameworks.

**Weaknesses:**

The paper is extremely dense, introducing very sophisticated and complex ideas that are usually given chapter-length expositions in just a few paragraphs, using very compact notation. Therefore it requires both a substantial amount of familiarity with the related literature and a lot of effort from the reader to understand all of it. Ideally there would have been more simple examples such as Example 1 that highlight the main intuitions and illustrate all of the important concepts, making life easier for the reader. I suggest to the authors to write up a much longer journal version of this paper that simply lays out the same content but with a lot more handholding for the reader, both in terms of conceptual clarifications as in concrete examples.

**Questions:**

Main questions:

1: As the authors point out, in the literature on abstractions it is common to add a set of allowed interventions to an SCM, bringing them closer to RCMs. In fact, this makes me wonder: is an SCM that includes a set of allowed interventions, simply an RCM which satisfies effectiveness, composition, and reversibility? Clarifying this would help in understanding the intuitions behind some of the results.

2: Relatedly, an RCM such as R' in Proposition 1 that satisfies these three axioms and has no proper extension, must be one which includes every intervention, and thus would simply be equivalent to an SCM, right? This would also help in interpreting the comment in 182: an RCM generalizes an SCM by not requiring all interventions to be allowed, and by not requiring the same axioms to hold. It might be interesting to compare how GSEMS fit into this picture. (GSEMS are Generalized Structural Equations Models, introduced by Halpern and Peters.)

3: Again a follow-up: L in Proposition 2 would then be equivalent to an SCM, right? It has no proper extension, thus it must include all interventions, and hence by Proposition 1 it satisfies the axioms for an SCM.

4: An obvious question: have you looked into whether new interesting results emerge if you generalize beyond constructive abstractions?

Minor technical questions/clarifications:
1: When introducing SCMs, why assume that U_V and Pa_V are strict subsets?

2: 228: "close under constructive abstraction".
This wasn't clear to me, because for effective we go from H to L, whereas for representable we go from L to H, so it sounds like close means something different in each case.

3: monotonicity: Here the variables are given a specific interpretation, right? X is treatment, and Z is prescribed treatment? Or is the idea that this relation holds for all potential outcomes? More generally, my background in PO is limited, and I have no sense of what this condition is capturing. Some intuition would be nice.

4: Theorem 2: I’m confused, here because members of S are part of the base language and thus do not include quantification, so are we assuming an implicit universal quantification? Because that sounds at odds with the possibility of using an existential quantifier in Definition 11.

Typos:

35: a "a...

44: tactictly



**Limitations:**

yes

---

> ### Author Rebuttal · Authors · 2023-08-10
>
> Thank you for the detailed review!
>
> Responses to questions:
>
> 1. Yes, fixing a set of allowed interventions over an SCM yields an RCM with these three properties. However an RCM with these properties does not always come from an SCM (see Ex. 3). We have clarified this on line 181, changing "possibly not sufficient" to "not sufficient, in light of Ex. 3" and added a footnote to the text of the example.
>
> 2. Indeed, such an RCM can be considered equivalent to an SCM.  It will, for example, be identical with regard to interpreting our probabilistic logical language. Note however that RCMs and SCMs (Def. 1, 3) are still defined as very different objects. We have updated the explanation on line 185 ("However, ...") to point out this equivalence explicitly.
> We are also grateful for the suggestion to compare Peters and Halpern's GSEMs, which were also observed to be expressively equivalent to Blom et al.'s "causal constraint models" (UAI 2019). These are similarly offered as "mechanism-free" generalizations of SCMs. But the focus in that work is on allowing multiple possible outcomes, corresponding, e.g., to multiple equilibria in a dynamical systems model. We've added citations to this important related work.
>
> 3. Yes, such an $\mathcal{L}$ would likewise be equivalent to an SCM.
>
> 4. We thank the reviewer for this suggestion, which we had not considered. Generalizing beyond constructive abstractions is an interesting direction for future work, as it may allow one to require, e.g. that $\mathcal{R}_{\mathrm{L}}$ be representable by an SCM with a specific graph.
>
> Minor questions:
>
> 1. We have changed the notation $\subset$ to $\subseteq$. Indeed, this was not intended to indicate a strict subset.
> 2. In both directions, the abstraction is from $\mathcal{L}$ to $\mathcal{H}$ (finer to coarser); the phrasing was in opposite orders though so we can see that it may have been misleading. Note that we have actually decided to remove the final claim of Prop. 2 (see response to Question 2 of Reviewer w4ki), so in the event, there will no longer be any conflict here.
> 3. This principle means that there are no units that do the opposite of what they were prescribed. To make this clearer, on line 283 we have added an explanation of the meaning of $X$, $Z$, which are indeed interpreted as treatment and prescribed treatment respectively, and elaborated on our explanation of monotonicity ("no defiers," line 281).
> 4. $S$ is meant to be a set of quantified (over $u$) assumptions in $\mathcal{L}_{\text{base}}$, making it consistent with Def. 11, although as you point out this was not explicit. We have clarified this in line 290.

---

> > ### Comment · Reviewer_Ef3C · 2023-08-11
> >
> > I have read the rebuttal of the authors. Thanks for the clarifications.

---

### Official Review · Reviewer_w4ki · 2023-07-26

**Soundness:** 4 excellent
**Presentation:** 2 fair
**Contribution:** 2 fair
**Rating:** 6
**Confidence:** 3

**Summary:**

This paper compares the Rubin causal model (RCM) and structural causal model (SCM) frameworks for causal inference. Specifically, the authors show that RCMs, when encoded with the composition and reversibility properties, represent the same space of counterfactual distributions as SCMs. Moreover, they show that all RCMs are constructive abstractions of some RCM that is representable by an SCM. Finally, they characterize the axioms and assumptions that are sound and complete for counterfactual inferences in both frameworks.

**Strengths:**

1. The paper offers a refreshing and productive perspective of the two causal inference frameworks that compares them on their axiomatic properties and abilities to encode causal assumptions and perform inferences, as opposed to more controversial philosophical aspects.


2. The work is logically grounded given the definitions and axioms and provides concrete connections between the frameworks. The logic is explained clearly through examples.


3. I found the usage of causal abstractions in the paper unexpected and very interesting, especially since the original sources were developed under the SCM framework. Causal abstractions provide a powerful framework for describing causal properties across different sets of variables. Prop. 2 demonstrates that certain properties such as effectiveness are preserved across abstractions, allowing for a more flexible way of interpreting RCMs.


4. Causal assumptions are an important aspect of causal inference research, but they are typically studied with respect to a fixed framework. This paper provides an interesting approach to encoding the assumptions in a manner that allows comparisons between the RCM and SCM frameworks.


**Weaknesses:**

1. It is not clear what is the ultimate takeaway of this work, as most of the paper details mathematical connections but not much about their implications. For example, Sec. 1.1 and Prop. 1 claim that SCMs and RCMs are equivalent frameworks in theory, but how should a reader change the way they approach causal inference given this knowledge?


2. The flow between different sections is somewhat strange. It is not clear exactly what the abstractions of Sec. 1.2 are adding to, say, the results of Prop 1. Sec. 2 also seems to be mostly unrelated to the previous section. Perhaps the transitions between the sections could be reworked to fix this.


3. As mentioned in the conclusion, Thm. 3 only applies to graphs for which the components connected through bidirected edges are complete cliques, meaning that the theory is currently incomplete for SCMs. This did not impact my score in this review, but this is definitely an interesting direction of future work. I am curious if the authors have any counterexamples of cases with other graphs where there are additional inequality constraints not captured by the given axioms.


I would also appreciate it if the authors could answer some of my questions in the next section.


**Questions:**

1. The preliminaries focus on the space of effective RCMs $R_{\text{eff}}$ and the space of SCMs $M_{\text{uniq}}$. Are there SCMs that are not in $M_{\text{uniq}}$? Is this uniqueness property a characterization of effectiveness in the SCM framework?


2. Can you clarify the relationship between Thm. 1 and Prop. 2? I think both results are interesting, but they seem to have conflicting messages. Prop. 2 seems to be saying that $\mathcal{H}$ is representable if it is an abstraction of a representable RCM, while Thm. 1 seems to be saying that all RCMs are abstractions of a representable RCM.


3. Is Eq. 6 reversed? It seems like the $z^+$ and $z^-$ should be swapped.


**Limitations:**

Limitations are clearly stated.

---

> ### Author Rebuttal · Authors · 2023-08-10
>
> Thank you for the constructive comments.
>
> In response to your very helpful questions:
>
> 1. We have chosen to focus on effective RCMs because this assumption is almost always made in practice. Effectiveness is so desirable that (as we have noted) one would go so far as to introduce additional variables should it fail to hold.
> On the SCM side, we focus on $M_{\mathrm{uniq}}$ because it is the most general class (up to measure zero) in which the SCM induces a unique counterfactual distribution, thus permitting comparison with that induced by an RCM. We have added a sentence explicating this. There are certainly SCMs that are not in $M_{\mathrm{uniq}}$. Given the above consideration, they are outside the scope of our paper, but one example is the following: $f_X(Y = y, U = u) = y$, $f_Y(X = x, U = u) = x$. If $X$, $Y$ are both binary then $(X, Y) = (0, 0), (1, 1)$ are both solutions to this SCM under any $u$.
> There is no relationship per se between the uniqueness property and effectiveness, but because of the interventional semantics effectiveness will hold in every SCM solution (whether these are unique or not), which we have also pointed out in a new footnote.
>
> 2. Spurred on by your comment, we have decided to remove the final claim of Prop. 2 (and its mention in Ex. 1) from the main text. We decided it is more important to focus attention on Thm. 1 and avoid the potential confusion you identified. Thm. 1 is central to the paper as a whole, as it identifies a source of apparent incompatibilities between the RCM and SCM frameworks—namely, the level of abstraction.
>
> 3. Indeed, $z^+$ and $z^-$ should be swapped in equation (6), and we have done this for the final version. We thank the reviewer for the catch.

---

> > ### Comment · Reviewer_w4ki · 2023-08-14
> > **RE: Rebuttal by Authors**
> >
> > I have read the rebuttal, and I thank the authors for answering my questions. I will maintain my positive rating.

---

### Official Review · Reviewer_vtiC · 2023-07-27

**Soundness:** 3 good
**Presentation:** 2 fair
**Contribution:** 4 excellent
**Rating:** 4
**Confidence:** 3

**Summary:**

The paper presents a logical framework to represent both the Rubin potential outcomes (PO) approach and the structural causal model (SCM) approach to causality. It shows that under mild assumption (composition and reversibility) every PO model is representable by an SCM. The paper proceeds to show how the underlying logical framework can then be used to elucidate the assumptions necessary for instrumental variable inference (commonly discussed in the PO context), in particular for the derivation of the local average treatment effect (LATE), and similarly, how the logical language permits the derivation of graphical conditions used for identifiability results in the SCM framework.

**Strengths:**

-- offers formal connection between potential outcome framework and structural causal model framework which elsewhere is often only stated informally
-- gives specific examples using the underlying logical language (developed in more detail in the papers cited) that show the need for specific assumptions in the PO framework that have been subject to much discussion
-- shows with examples how to recover identifiability results in the graphical models framework
-- offers the prospect of a unifying logical framework for understanding causality

These are very interesting results and I really encourage a thorough and clear discussion of how they fit in.

**Weaknesses:**

-- Section 1 provides the most interesting contribution by formally connecting the PO and SCM framework, but the paper then quickly moves on to treat questions of causal inference, rather than fully explaining what can and cannot be translated between the two frameworks now. There has been a very substantive discussion in the literature about (a) the connection between PO and SCMs and to what extent it has been addressed by the single world intervention graphs (SWIGs, citation 28) and (b) the differences in the treatment of counterfactuals even within different versions of the structural equation modeling framework. The present paper is missing a more detailed discussion of how the present contributions fit into that context. Obviously, this would require more space and so a journal paper would be more appropriate for this material. As it stands the paper is very dense and provides a mixture of half a discussion of the unification of PO and SCM and then a couple of very nice examples of re-derivations of known results using the logical framework.

**Questions:**

1) I would very much like to understand the connection of the present work to SWIGs. This seems to me to be crucial for a paper that provides a unifying approach to PO and SCM frameworks.
2) Am I missing something about the causal abstraction? A whole page is dedicated to it, but for what is discussed in the actual paper it appears to only ensure that one can appropriately control the state spaces of the variables.
3) Please explain cf-sep as a generalization of d-separation.

**Limitations:**

limitations are appropriately discussed

---

> ### Author Rebuttal · Authors · 2023-08-10
>
> We thank the reviewer for their insightful comments!
>
> We very much appreciate the point about SWIGs and alternative frameworks for reasoning about counterfactuals. In fact, in an earlier draft we had included some remarks about SWIGs in particular, including how they fit into our logical framework. We removed this due to space constraints, but we could use a fraction of the additional alloted page to discuss that connection (still relegating some of the more technical discussion to the appendix).
>
> In short, we can define FFRCISTGs as a subclass of RCMs and show that SWIGs are in fact sound and complete (in the logical sense) with respect to FFRCISTGs. Specifically, the SWIG framework derives the crucial independences and restrictions of these models within our formal language L. We will explain this connection—also re-emphasizing its importance in the larger reconciliation project between counterfactual and graphical traditions—in a new, brief subsection (2.3).
>
> There are aspects of the SWIG framework that are, so to speak, extralogical—meaning that they pertain to how easy it is for humans to understand and assess them—but we believe it is a further testament to the framework in the present work that it can help locate these "quasi-graphical" models in the broader context of RCMs and SCMs, particularly from an inferential perspective.
>
> In response to Question 2 about abstraction, we have edited the text to make the role of abstraction maximally clear. The point here is that the state space of the RCM is given, and our Thm. 1 reveals that by moving to some finer state space (given in the proof) it is possible to ensure that principles characteristic of SCMs hold while maintaining consistency between the finer and coarser spaces.
>
> As for Question 3 about $\textsf{cf-sep}$, it is a generalization in the sense that $\textsf{cf-sep}$ and exclusion restrictions ($\textsf{ER}^{\mathcal{G}}$) suffice (by none other than our completeness result) to derive *all* conditional independences given by d-separation. On the other hand, every instance of $\textsf{cf-sep}$ represents *some* conditional independence implied by d-separation (in an appropriate twin network); our result just shows that strictly speaking the others can be derived from these few (given $\textsf{ER}$). We have elaborated the discussion in our paper to make this relationship clearer.

---

> > ### Comment · Reviewer_vtiC · 2023-08-13
> > **great work, but would like to see connection to SWIGs first before recommending acceptance**
> >
> > I have read all the comments and thank the authors for the very clear and helpful feedback. I think this is a very interesting paper, but would like to see the details of the connection to the SWIGs worked out before recommending acceptance. I find the rebuttal comments to my review most intriguing and would like to understand them fully. I like this work a lot and think it will eventually be a very significant contribution once the the connections are discussed and clarified. I will maintain my rating given the current manuscript.

---

> > > ### Author Response · Authors · 2023-08-15
> > > **Further details**
> > >
> > > We are very grateful to the reviewer for the positive and encouraging remarks, and we completely understand the reservations about adding new material. Just in case it might be helpful, we wanted to offer some further clarifying remarks about the role of SWIGs and what we plan to include about them.
> > >
> > > We see the SWIG framework not so much as a means for comparing the RCM and SCM approaches, but rather as an elegant example of how ideas and concepts from the two approaches can be productively combined. As such, it fits nicely in the section on inference, highlighting methods that draw from both approaches.
> > >
> > > A SWIG can be understood as a kind of graphical model, characterized in terms of factorization of joint distributions on potential outcomes. Aside from facilitating reasoning with a wider class of expressions than is possible with the notation of do-calculus — see [1] on (conditional) path-specific effects for a great example of this — it also enjoys completeness with respect to independencies implied by a particular kind of RCM (namely, a FFRCISTG [3]). Drawing on existing results [2], we can re-present (and ever so slightly strengthen) this fact as a completeness result for the language L in our paper. The point of doing so is again to gauge the strength of assumptions implied by a SWIG in a common language, which we see as very much in the spirit of the SWIG framework.
> > >
> > > In sum, we wanted to clarify that the new subsection is not intended to introduce fundamentally new results, but merely to illuminate how this graphical framework fits into the narrative of our paper. In addition to illustrating the benefits of hybrid frameworks, it also anticipates some of the very connections that we have tried to bring out in our own work. In any case, we would be delighted to answer any other questions about this, or about what we intend to include in an additional allotted page.
> > >
> > > [1] D. Malinsky, I. Shpitser, and T. S, Richardson. A potential outcomes calculus for identifying conditional path-specific effects. In K. Chaudhuri and M. Sugiyama, editors, Proceedings of the Twenty-Second International Conference on Artificial Intelligence and Statistics, volume 89, pages 3080–3088, 2019.
> > >
> > > [2]  T. S. Richardson and J. M. Robins. Single world intervention graphs (SWIGs): A unification of the counterfactual and graphical approaches to causality. Working Paper Number 128, Center for Statistics and the Social Sciences, University of Washington, 2013.
> > >
> > > [3] J. M. Robins. A new approach to causal inference in mortality studies with sustained exposure periods – applications to control of the healthy worker survivor effect. Mathematical Modeling 7, 1393–1512, 1986.

---

### Author Rebuttal · Authors · 2023-08-10

We are sincerely grateful to the reviewers for their truly helpful and constructive feedback, and also for their encouraging remarks about the work and its significance. We feel that addressing their constructive suggestions has improved the paper and made it more effective.

In the individual responses below we comment on the specific points raised by each. At a general level, we appreciate the concern from several of the reviewers about density. In addition to some of the particular amendments detailed in the individual responses, we are enthusiastic about using the extra page for the final version to add further explanation of key ideas, notation, and concepts. In addition, as discussed in the response to vtiC, we will use about 1/3 of the extra page to comment on how SWIGs fit into the broader framework, emphasizing the importance of such work for illuminating the many facets and connections between research programs in these two traditions.

We look forward to further discussion and opportunities for clarification, and want to thank the reviewers again for their careful and thoughtful attention to our work.

---

### Decision · Program_Chairs · 2023-09-21

**Decision:**

Accept (poster)

**Comment:**

This paper builds a bridge between the two principal schools of thought regarding causal models: potential outcomes and structural causal models. The reviewers were appreciative about the contributions made to the foundations of causality theory and the excellent quality of writing.